# ON THE SATURATION EFFECT OF KERNEL RIDGE REGRESSION

**Yicheng Li & Haobo Zhang**
Center for Statistical Science, Department of Industrial Engineering
Tsinghua University, Beijing, China
`{liyc22,zhang-hb21}@mails.tsinghua.edu.cn`

**Qian Lin**
Center for Statistical Science, Department of Industrial Engineering
Tsinghua University, Beijing, China
and
Beijing Academy of Artificial Intelligence, Beijing, China
`qianlin@tsinghua.edu.cn`

## ABSTRACT

The saturation effect refers to the phenomenon that the kernel ridge regression (KRR) fails to achieve the information theoretical lower bound when the smoothness of the underground truth function exceeds certain level. The saturation effect has been widely observed in practices and a saturation lower bound of KRR has been conjectured for decades. In this paper, we provide a proof of this long-standing conjecture.

## 1 INTRODUCTION

Suppose that we have observed $n$ i.i.d. samples $\{(x_i, y_i)\}_{i=1}^n$ from an unknown distribution $\rho$ supported on $\mathcal{X} \times \mathcal{Y}$ where $\mathcal{X} \subseteq \mathbb{R}^d$ and $\mathcal{Y} \subseteq \mathbb{R}$. One of the central problems in the statistical learning theory is to find a function $\hat{f}$ based on these observations such that the generalization error

$$\mathbb{E}_{(x,y)\sim\rho}\left[\left(\hat{f}(x) - y\right)^2\right] \tag{1}$$

is small. It is well known that the conditional mean $f_\rho^*(x) := \mathbb{E}_\rho[\, y \mid x \,] = \int_{\mathcal{Y}} y \mathrm{d}\rho(y|x)$ minimizes the square loss $\mathcal{E}(f) = \mathbb{E}_\rho(f(x)-y)^2$ where $\rho(y|x)$ is the distribution of $y$ conditioning on $x$. Thus, this question is equivalent to looking for an $\hat{f}$ such that the generalization error

$$\mathbb{E}_{x\sim\mu}\left[\left(\hat{f}(x) - f_\rho^*(x)\right)^2\right] \tag{2}$$

is small, where $\mu$ is the marginal distribution of $\rho$ in $\mathcal{X}$. In other words, $\hat{f}$ can be viewed as an estimator of $f_\rho^*$. When there is no explicit parametric assumption made on the distribution $\rho$ or the function $f_\rho^*$, researchers often assumed that $f_\rho^*$ falls into a class of certain functions and developed lots of non-parametric methods to estimate $f_\rho^*$ (e.g., Györfi (2002); Tsybakov (2009)).

The kernel method, one of the most widely applied non-parametric regression methods (e.g., Kohler & Krzyzak (2001); Cucker & Smale (2001); Caponnetto & De Vito (2007); Steinwart et al. (2009); Fischer & Steinwart (2020)), assumes that $f_\rho^*$ belongs to certain reproducible kernel Hilbert space (RKHS) $\mathcal{H}$, a separable Hilbert space associated to a kernel function $k$ defined on $\mathcal{X}$. The kernel ridge regression (KRR), which is also known as the Tikhonov regularization or regularized least squares, estimates $f_\rho^*$ by solving the penalized least square problem:

$$\hat{f}_\lambda^{\mathrm{KRR}} = \arg\min_{f\in\mathcal{H}} \left(\frac{1}{n}\sum_{i=1}^n (y_i - f(x_i))^2 + \lambda\|f\|_{\mathcal{H}}^2\right), \tag{3}$$

where $\lambda > 0$ is the so-called regularization parameter. By the representer theorem (see e.g., Andreas Christmann (2008)), this estimator has an explicit formula (please see (8) for the exact meaning of the notation):

$$\hat{f}_\lambda^{\mathrm{KRR}}(x) = \mathbb{K}(x, X) \left( \mathbb{K}(X, X) + n\lambda I \right)^{-1} \mathbf{y}.$$

Theories have been developed for KRR from many aspects over the last decades, especially for the convergence rate of the generalization error. For example, if $f_\rho^* \in \mathcal{H}$ without any further smoothness assumptions, Caponnetto & De Vito (2007) and Steinwart et al. (2009) showed that the generalization error of KRR achieves the information theoretical lower bound $n^{-\frac{1}{1+\beta}}$, where $\beta$ is a characterizing quantity of the RKHS $\mathcal{H}$ (see e.g., the eigenvalue decay rate defined in Condition **(A)**).

Further studies reveal that when more regularity(or smoothness) of $f_\rho^*$ is assumed, the KRR fails to achieve the information theoretic lower bound of the generalization error. More precisely, when $f_\rho^*$ is assumed to belong some interpolation space $[\mathcal{H}]^\alpha$ of the RKHS $\mathcal{H}$ where $\alpha > 2$, the information theoretical lower bound of the generalization error is $n^{-\frac{\alpha}{\alpha+\beta}}$ (Rastogi & Sampath, 2017) and the best upper bound of the generalization error of KRR is $n^{-\frac{2}{2+\beta}}$ (Caponnetto & De Vito, 2007). This gap between the best existing KRR upper bounds and the information theoretical lower bounds of the generalization error has been widely observed in practices (e.g. Bauer et al. (2007); Gerfo et al. (2008)). It has been conjectured for decades that no matter how carefully one tunes the KRR, the rate of the generalization error can not be faster than $n^{-\frac{2}{2+\beta}}$ (Gerfo et al., 2008; Dicker et al., 2017). This phenomenon is often referred to as the saturation effect (Bauer et al., 2007) and we refer to the conjectural fastest generalization error rate $n^{-\frac{2}{2+\beta}}$ of KRR as the saturation lower bound. The main focus of this paper is to prove this long-standing conjecture.

## 1.1 RELATED WORK

KRR also belongs to the spectral regularization algorithms, a large class of kernel regression algorithms including kernel gradient descent, spectral cut-off, etc, see e.g. Rosasco et al. (2005); Bauer et al. (2007); Gerfo et al. (2008); Mendelson & Neeman (2010). The spectral regularization algorithms were originally proposed to solve the linear inverse problems (Engl et al., 1996), where the saturation effect was firstly observed and studied (Neubauer, 1997; Mathé, 2004; Herdman et al., 2010). Since the spectral algorithms were introduced into the statistical learning theory, the saturation effect has been also observed in practice and reported in literatures (Bauer et al., 2007; Gerfo et al., 2008).

Researches on spectral algorithms show that the asymptotic performance of spectral algorithms is mainly determined by two ingredients (Bauer et al., 2007; Rastogi & Sampath, 2017; Blanchard & Mücke, 2018; Lin et al., 2018). One is the relative smoothness(regularity) of the regression function with respect to the kernel, which is also referred to as the source condition (see, e.g. Bauer et al. (2007, Section 2.3)). The other is the qualification of the spectral algorithm, a quantity describing the algorithm's fitting capability(see, e.g. Bauer et al. (2007, Definition 1)). It is widely believed that algorithms with low qualification can not achieve the information theoretical lower bound when the regularity of $f_\rho^*$ is high. This is the (conjectural) saturation effect for the spectral regularized algorithms (Bauer et al., 2007; Lin & Cevher, 2020; Lian et al., 2021). To the best of our knowledge, most works pursue showing that spectral regularized algorithm with high qualification can achieve better generalization error rate while few work tries to answer this conjecture directly (Gerfo et al., 2008; Dicker et al., 2017).

The main focus of this paper is to provide a rigorous proof of the saturation effect of KRR for its simplicity and popularity. The technical tools introduced here might help us to solve the saturation effect of other spectral algorithms.

**Notation.** Let us denote by $X = (x_1, \ldots, x_n)$ the sample input matrix and $\mathbf{y} = (y_1, \ldots, y_n)'$ the sample output vector. We denote by $\mu$ the marginal distribution of $\rho$ on $\mathcal{X}$. Let $\epsilon_i := y_i - f^*(x_i)$ be the noise.

We use $L^p(\mathcal{X}, \mathrm{d}\mu)$ (sometimes abbreviated as $L^p$) to represent the Lebesgue $L^p$ spaces, where the corresponding norm is denoted by $\|\cdot\|_{L^p}$. Hence, we can express the generalization error as

$$\mathbb{E}_{x \sim \mu}\left[\left(\hat{f}(x) - f_\rho^*(x)\right)^2\right] = \left\|\hat{f} - f_\rho^*\right\|_{L^2}^2.$$

We use the asymptotic notations $O(\cdot)$, $o(\cdot)$, $\Omega(\cdot)$ and $\omega(\cdot)$. We also denote $a_n \asymp b_n$ iff $a_n = O(b_n)$ and $a_n = \Omega(b_n)$. We use the asymptotic notations in probability $O_\mathbb{P}(\cdot)$ and $\Omega_\mathbb{P}(\cdot)$ to state our results. Let $\{a_n\}_{n \geq 1}$ be a sequence of positive numbers and $\{\xi_n\}_{n \geq 1}$ a sequence of non-negative random variables. If for any $\delta > 0$ there exists $M_\delta > 0$ and $N_\delta > 0$ such that $\mathbb{P}\{\xi_n < M_\delta a_n\} \geq 1 - \delta$, $\forall n \geq N_\delta$, we say that $\xi_n$ is bounded (above) by $a_n$ in probability and write $\xi_n = O_\mathbb{P}(a_n)$. The definition of $\Omega_\mathbb{P}(b_n)$ follows similarly.

## 2 BRIEF REVIEW OF THE SATURATION EFFECT

### 2.1 REGRESSION OVER REPRODUCING KERNEL HILBERT SPACE

Throughout the paper, we assume that $\mathcal{X} \subset \mathbb{R}^d$ is compact and $k$ is a continuous positive-definite kernel function defined on $\mathcal{X}$. Let $T : L^2(\mathcal{X}, \mathrm{d}\mu) \to L^2(\mathcal{X}, \mathrm{d}\mu)$ be the integral operator defined by

$$(Tf)(x) := \int_\mathcal{X} k(x, y)f(y)\mathrm{d}\mu(y). \tag{4}$$

It is well known that $T$ is trace-class(thus compact), positive and self-adjoint (Steinwart & Scovel, 2012). The spectral theorem of compact self-adjoint operators together with Mercer's theorem (see e.g., Steinwart & Scovel (2012)) yield that

$$T = \sum_{i \in N} \lambda_i \langle \cdot, e_i \rangle_{L^2} e_i, \tag{5}$$

$$k(x, y) = \sum_{i \in N} \lambda_i e_i(x)e_i(y), \tag{6}$$

where $\lambda_i$'s are the positive eigenvalues of $T$ in descending order, $e_i$'s are the corresponding eigenfunctions, and $N \subseteq \mathbb{N}$ is an at most countable index set. Let $\mathcal{H}$ be the separable RKHS associated to the kernel $k$ (see, e.g., Wainwright (2019, Chapter 12)). One may easily verify that $\left\{\lambda_i^{1/2} e_i\right\}_{i \in N}$ is an orthonormal basis of $\mathcal{H}$. Since we are interested in the infinite-dimensional cases, we may assume that $N = \mathbb{N}$.

Recall that the kernel ridge regression estimates the regression function $f_\rho^*$ through the following optimization problem:

$$\hat{f}_\lambda^{\mathrm{KRR}} = \underset{f \in \mathcal{H}}{\arg\min} \left(\frac{1}{n}\sum_{i=1}^n (y_i - f(x_i))^2 + \lambda\|f\|_\mathcal{H}^2\right). \tag{7}$$

The representer theorem (see e.g., Andreas Christmann (2008)) implies that

$$\hat{f}_\lambda^{\mathrm{KRR}}(x) = \mathbb{K}(x, X)(\mathbb{K}(X, X) + n\lambda I)^{-1}\mathbf{y}, \tag{8}$$

where

$$\mathbb{K}(x, X) = (k(x, x_1), \ldots, k(x, x_n)) \text{ and } \mathbb{K}(X, X) = \left(k(x_i, x_j)\right)_{n \times n}.$$

The following conditions are commonly adopted when discussing the performance of $\hat{f}_\lambda^{\mathrm{KRR}}$.

(**A**) *Eigenvalue decay rate: there exists absolute constant $c_1 > 0$, $c_2 > 0$ and $\beta \in (0, 1)$ such that the eigenvalues $\lambda_i's$ of $T$, the integral operator associated to the kernel $k$, satisfy*

$$c_1 i^{-1/\beta} \leq \lambda_i \leq c_2 i^{-1/\beta}, \, \forall \, i = 1, 2, \ldots \tag{9}$$

(**B**) *Smoothness condition: the conditional mean $f_\rho^* = \mathbb{E}_\rho[\, y \mid x \,]$ satisfies that $\left\|f_\rho^*\right\|_\mathcal{H} \leq R$ for some $R > 0$;*

(**C**) *Moment condition of the noise:*

$$\int_{\mathcal{Y}} |y - f_\rho^*(x)|^m \, \mathrm{d}\rho(y|x) \leq \frac{1}{2} m! \sigma^2 M^{m-2}, \quad \mu\text{-a.e. } x \in \mathcal{X}, \quad \forall m = 2, 3, \ldots,$$

*where $\sigma, M > 0$ are some constants.*

The quantity $\beta$ appearing in the condition (**A**) is often referred to as the eigenvalue decay rate of an RKHS or the corresponding kernel function $k$. It describes the span of the RKHS and depends only on the kernel function $k$ (or equivalently, the RKHS $\mathcal{H}$). This polynomial decay rate condition is quite standard in the literature and is also known as the capacity condition or effective dimension condition (Caponnetto & De Vito, 2007; Steinwart et al., 2009; Blanchard & Mücke, 2018), which is closely related to the covering/entropy number conditions of the RKHS (see, e.g., Steinwart et al. (2009, Theorem 15)). The condition (**B**) requires that the conditional mean $f_\rho^*(x)$ falls into the RKHS $\mathcal{H}$ with norm smaller than a given constant $R$. The condition (**C**) requires that the tail probability of the 'noise' decays fast, which is satisfied if the noise $\varepsilon = y - f_\rho^*(x)$ is bounded or sub-Gaussian.

**Proposition 2.1** (Optimality of KRR). *Suppose that $\mathcal{H}$ satisfies the condition (**A**) and $\mathcal{P}$ consists of all the distributions satisfying the conditions (**B**) and (**C**).*

*i) The minimax rate of estimating $f_\rho^*$ is $n^{-\frac{1}{1+\beta}}$, i.e., we have*

$$\inf_{\hat{f}} \sup_{\rho \in \mathcal{P}} \mathbb{E}_\rho \left\| \hat{f} - f_\rho^* \right\|_{L^2}^2 = \Omega\left( n^{-\frac{1}{1+\beta}} \right), \tag{10}$$

*where $\inf_{\hat{f}}$ is taken over all estimators and both the expectation $\mathbb{E}_\rho$ and the conditional mean $f_\rho^*$ depend on $\rho$.*

*ii) If we choose $\lambda \asymp n^{-\frac{1}{1+\beta}}$, then we have*

$$\left\| \hat{f}_\lambda^{\mathrm{KRR}} - f_\rho^* \right\|_{L^2}^2 = O_\mathbb{P}\left( n^{-\frac{1}{1+\beta}} \right). \tag{11}$$

This theorem comes from a combination (with a slight modification) of the upper rates and lower rates given in Caponnetto & De Vito (2007). It says that the optimally tuned KRR can achieve the information theoretical lower bound if $f_\rho^*(x)$ falls into $\mathcal{H}$ and no further regularity condition is imposed.

## 2.2 THE SATURATION EFFECT

When further regularity assumption is made on the conditional mean $f_\rho^*(x)$, there will be a gap between the information theoretical lower bound and the upper bound provided by the KRR. This phenomenon is now referred to the saturation effect in KRR. In order to explicitly describe the saturation effect, we need to introduce a family of interpolation spaces of the RKHS $\mathcal{H}$ (see, e.g. Fischer & Steinwart (2020)).

For any $s \geq 0$, the operator $T^s : L^2(\mathcal{X}) \to L^2(\mathcal{X})$ is given by

$$T^s(f) = \sum_{i \in N} \lambda_i^s \langle f, e_i \rangle_{L^2} e_i \tag{12}$$

and the interpolation space $[\mathcal{H}]^\alpha$ for $\alpha \geq 0$ is defined by

$$[\mathcal{H}]^\alpha := \mathrm{Ran}\, T^{\alpha/2} = \left\{ \sum_{i \in N} a_i \lambda_i^{\alpha/2} e_i \middle| (a_i)_{i \in N} \in \ell^2(N) \right\}, \tag{13}$$

with the inner product defined by

$$\langle f, g \rangle_{[\mathcal{H}]^\alpha} = \left\langle T^{-\alpha/2} f, T^{-\alpha/2} g \right\rangle_{L^2}. \tag{14}$$

From the definition, it is easy to verify that $\left\{ \lambda_i^{\alpha/2} e_i \right\}_{i \in N}$ forms an orthonormal basis of $[\mathcal{H}]^\alpha$ and thus $[\mathcal{H}]^\alpha$ is also a separable Hilbert space. It is obvious that $[\mathcal{H}]^0 \subseteq L^2$ and $[\mathcal{H}]^1 \subseteq \mathcal{H}$.

Moreover, we have isometric isomorphisms $T^{s/2} : [\mathcal{H}]^\alpha \to [\mathcal{H}]^{\alpha+s}$ and compact embeddings $[\mathcal{H}]^{\alpha_1} \hookrightarrow [\mathcal{H}]^{\alpha_2}, \quad \forall \alpha_1 > \alpha_2 \geq 0$. The example below illustrates the intuition of $\alpha$: it describes the smoothness of functions with respect to the kernel.

**Example 2.1** (Sobolev RKHS (Fischer & Steinwart, 2020))**.** Let $\mathcal{X} = \Omega \subseteq \mathbb{R}^d$ be a bounded domain with smooth boundary. We consider the Sobolev space $\mathcal{H} = H^s(\Omega)$, which, roughly speaking, consists of functions with weak derivatives up to order $s$, see, e.g., Adams & Fournier (2003). It is known that if $s > \frac{d}{2}$, we have the Sobolev embedding $H^s(\Omega) \hookrightarrow C^r(\Omega)$ where $C^r(\Omega)$ is the Hölder space of continuous differentiable functions and $r = s - \frac{d}{2}$, and thus $H^s(\Omega)$ is an RKHS (Fischer & Steinwart, 2020). Moreover, by the method of real interpolation (Steinwart & Scovel, 2012, Theorem 4.6), we have $[\mathcal{H}]^\alpha \cong H^{\alpha s}(\Omega)$ for $\alpha > 0$. This example shows that the larger the $\alpha$, the "smoother" the functions in $[\mathcal{H}]^\alpha$.

If one believes that $f_\rho^*(x)$ possesses more regularity (i.e., $f_\rho^*(x) \in [\mathcal{H}]^\alpha$), we may replace the condition (**B**) by the following condition:

(**B′**) Smoothness condition: the conditional mean $f_\rho^* = \mathbb{E}_\rho[\, y \mid x \,]$ satisfies that $\left\| f_\rho^* \right\|_{[\mathcal{H}]^\alpha} \leq R$ for some $R > 0$.

In this condition, the parameter $\alpha$ describes the smoothness of the regression function with respect to the underlying kernel. The larger $\alpha$ is, the "smoother" the regression function is. This assumption is also referred to as the source condition in the literature, see, e.g., Bauer et al. (2007); Rastogi & Sampath (2017). With this new regularity assumption, we have the following statement.

**Proposition 2.2** (Saturation phenomenon of KRR)**.** *Suppose that $\mathcal{H}$ satisfies the condition (**A**) and $\mathcal{P}$ consists of all the distributions satisfying the conditions (**B′**) and (**C**).*

**i)** *The minimax rate of estimating $f_\rho^*$ is $n^{-\frac{\alpha}{\alpha+\beta}}$, i.e., we have*

$$\inf_{\hat{f}} \sup_{\rho \in \mathcal{P}} \mathbb{E}_\rho \left\| \hat{f} - f_\rho^* \right\|_{L^2}^2 = \Omega\left( n^{-\frac{\alpha}{\alpha+\beta}} \right), \tag{15}$$

*where $\inf_{\hat{f}}$ is taken over all estimators and both the expectation $\mathbb{E}_\rho$ and the regression function $f_\rho^*$ depend on $\rho$.*

**ii)** *Let $\tilde{\alpha} = \min(\alpha, 2)$. Then, by choosing $\lambda \asymp n^{-\frac{1}{\tilde{\alpha}+\beta}}$, we have*

$$\left\| \hat{f}_\lambda^{\mathrm{KRR}} - f^* \right\|_{L^2}^2 = O_{\mathbb{P}}\left( n^{-\frac{\tilde{\alpha}}{\tilde{\alpha}+\beta}} \right). \tag{16}$$

This proposition comes from a combination (with a slight modification) of the lower rate derived in Rastogi & Sampath (2017, Corollary 3.3) and the upper rate given in Fischer & Steinwart (2020, Theorem 1 (ii)). It says that the optimally tuned KRR can achieve the information theoretical lower bound if $f_\rho^*(x) \in [\mathcal{H}]^\alpha$ when $\alpha \leq 2$. However, when $\alpha > 2$, there is a gap between the information theoretical lower bound and the upper bound provided by the KRR.

## 3 MAIN RESULTS

We introduce two additional assumptions in order to state our main result.

**Assumption 1** (RKHS)**.** We assume that $\mathcal{H}$ is an RKHS over a compact set $\mathcal{X} \subseteq \mathbb{R}^d$ associated with a Hölder-continuous kernel $k$, that is, there exists some $s \in (0, 1]$ and $L > 0$ such that

$$|k(x_1, x_2) - k(y_1, y_2)| \leq L \|(x_1, x_2) - (y_1, y_2)\|_{\mathbb{R}^{d \times d}}^s, \quad \forall x_1, x_2, y_1, y_2 \in \mathcal{X}.$$

**Assumption 2** (Noise)**.** The conditional variance satisfies that

$$\mathbb{E}_{(x,y) \sim \rho}\left[ (y - f_\rho^*(x))^2 \mid x \right] \geq \bar{\sigma}^2 > 0, \quad \mu\text{-a.e. } x \in \mathcal{X}. \tag{17}$$

The first assumption is a Hölder condition on the kernel, which is slightly stronger than assuming that $k$ is continuous. It is satisfied, for example, when $k$ is Lipschitz or $C^1$. Kernels that satisfy this assumption include the popular RBF kernel, Laplace kernel and kernels of the form $(1 - \|x - y\|)_+^p$

(Wendland, 2004, Theorem 6.20), and kernels associated with Sobolev RKHS introduced in Example 2.1.

The second assumption requires that the variance of the noise $\varepsilon = y - f_\rho^*(x)$ is lower bounded. When $y = f^*(x) + \varepsilon$ where $\varepsilon \sim N(0, \sigma^2)$ is an independent noise, the second assumption simply requires that $\sigma \neq 0$. In other words, Assumption 2 is a fairly weak assumption which just requires that the noise is non-vanishing almost everywhere.

Now we are ready to state our main theorem.

**Theorem 3.1** (Saturation effect). *Suppose that $\mathcal{H}$ satisfies the condition* (**A**), *the distribution $\rho$ satisfies that $f_\rho^* \neq 0$ and $f_\rho^* \in [\mathcal{H}]^\alpha$ for some $\alpha \geq 2$, and Assumptions 1 and 2 hold.*

*For any $\delta > 0$, for any choice of regularization parameter $\lambda(n)$ satisfying that $\lambda(n) \to 0$, we have that, for sufficiently large $n$,*

$$\mathbb{E}\left[\left\|\hat{f}_\lambda^{\mathrm{KRR}} - f^*\right\|_{L^2}^2 \;\Big|\; X\right] \geq cn^{-\frac{2}{2+\beta}} \tag{18}$$

*holds with probability at least $1 - \delta$ for some positive constant c. Consequently, we have*

$$\mathbb{E}\left[\left\|\hat{f}_\lambda^{\mathrm{KRR}} - f^*\right\|_{L^2}^2 \;\Big|\; X\right] = \Omega_{\mathbb{P}}\left(n^{-\frac{2}{2+\beta}}\right).$$

**Remark 3.2.** The saturation effect of KRR states that when the regression function is very smooth, i.e., $\alpha \geq 2$, no matter how the regularization parameter is tuned, the convergence rate of KRR is bounded below by $\frac{2}{2+\beta}$. The saturation lower bound also coincides with the upper bound (16) in Proposition 2.2. Therefore, Theorem 3.1 rigorously proves the saturation effect of KRR.

Moreover, we would like to emphasize that the saturation lower bound is established for arbitrary fixed non-zero $f^* \in [\mathcal{H}]^\alpha$, and it is essentially different from the information theoretical lower bound, e.g., (15), in both the statement and the proof technique.

### 3.1 SKETCH OF THE PROOF

We present the sketch of our proofs in this part and defer the complete proof to Section B. Let us introduce the sampling operator $K_x : \mathbb{R} \to \mathcal{H}$ defined by $K_x y = yk(x, \cdot)$ and its adjoint operator $K_x^* : \mathcal{H} \to \mathbb{R}$ given by $K_x^* f = f(x)$. We further introduce the sample covariance operator $T_X : \mathcal{H} \to \mathcal{H}$ by

$$T_X := \frac{1}{n} \sum_{i=1}^{n} K_{x_i} K_{x_i}^*, \tag{19}$$

and define the sample basis function

$$g_Z := \frac{1}{n} \sum_{i=1}^{n} K_{x_i} y_i \in \mathcal{H}. \tag{20}$$

The following explicit operator form of the solution of KRR is shown in Caponnetto & De Vito (2007):

$$\hat{f}_\lambda^{\mathrm{KRR}} = (T_X + \lambda)^{-1} g_Z. \tag{21}$$

The first step of our proof is the bias-variance decomposition, which differs from the commonly used approximation-estimation error decomposition in the literature (e.g. Caponnetto & De Vito (2007); Fischer & Steinwart (2020)). It can be shown that

$$\mathbb{E}\left[\left\|\hat{f}_\lambda^{\mathrm{KRR}} - f_\rho^*\right\|_{L^2}^2 \;\Big|\; X\right] = \lambda^2 \left\|(T_X + \lambda)^{-1} f_\rho^*\right\|_{L^2}^2 + \frac{1}{n^2} \sum_{i=1}^{n} \sigma_{x_i}^2 \left\|(T_X + \lambda)^{-1} k(x_i, \cdot)\right\|_{L^2}^2$$

$$=: \mathbf{Bias}^2 + \mathbf{Var}.$$

Then, the desired lower bound can be derived by proving the following lower bounds of the two terms respectively:

$$\textbf{Bias}^2 = \Omega_{\mathbb{P}}\left(\lambda^2\right), \quad \textbf{Var} = \Omega_{\mathbb{P}}\left(\frac{\lambda^{-\beta}}{n}\right). \tag{22}$$

These two lower bounds follow from our bias-variance trade-off intuition: smaller $\lambda$ (less regularization) leads to smaller bias but larger variance, while the variance decreases as $n$ increases. They also coincide with the main terms of the upper bound in the literature, see, e.g., Caponnetto & De Vito (2007); Fischer & Steinwart (2020).

**The bias term** First, we establish the approximation $\left\|(T_X + \lambda)^{-1} f_\rho^*\right\|_{L^2}^2 \approx \left\|(T + \lambda)^{-1} f_\rho^*\right\|_{L^2}^2$, where we refine the concentration result between $(T_X + \lambda)^{-1/2}$ and $(T + \lambda)^{-1/2}$ obtained in the previous literature. Second, we use the eigen-decomposition and the fact that KRR's qualification is limited to show that $\lambda \left\|(T + \lambda)^{-1} f_\rho^*\right\|_{L^2} \geq c\lambda$ for some constant $c > 0$. Consequently, we have

$$\textbf{Bias}^2 \approx \lambda^2 \left\|(T + \lambda)^{-1} f_\rho^*\right\|_{L^2}^2 \geq c\lambda^2.$$

This lower bound shows that the bais of KRR can only decrease in linear order with respect to $\lambda$ no matter how smooth the regression function is, limiting the performance of KRR.

**The variance term** We first rewrite the variance term in matrix forms and deduce that

$$\textbf{Var} \geq \frac{\bar{\sigma}^2}{n} \int_{\mathcal{X}} \left\|(T_X + \lambda)^{-1} k(x, \cdot)\right\|_{L^2, n}^2 \mathrm{d}\mu(x), \tag{23}$$

where $\|f\|_{L^2, n}^2 := \frac{1}{n} \sum_{i=1}^n f(x_i)^2$. This observation (23) is key and novel in the proof, allowing us to make the following two-step approximation:

$$\left\|(T_X + \lambda)^{-1} k(x, \cdot)\right\|_{L^2, n}^2 \approx \left\|(T + \lambda)^{-1} k(x, \cdot)\right\|_{L^2, n}^2 \approx \left\|(T + \lambda)^{-1} k(x, \cdot)\right\|_{L^2}^2.$$

The main difficulty here is to control the errors in the approximation so that they are infinitesimal compared to the main term, while errors of the same order as the main term are sufficient in the proof of upper bounds. To resolve the difficulty, we refine the analysis by combining both the integral operator technique (e.g. in Caponnetto & De Vito (2007)) and the empirical process technique (e.g. in Steinwart et al. (2009)), applying tight concentration inequalities and analyzing the covering number of regularized basis function family $\left\{(T + \lambda)^{-1} k(x, \cdot)\right\}_{x \in \mathcal{X}}$. Finally, by Mercer's theorem and the eigenvalue decay rate, we obtain that

$$\int_{\mathcal{X}} \left\|(T + \lambda)^{-1} k(x, \cdot)\right\|_{L^2}^2 \mathrm{d}\mu(x) = \int_{\mathcal{X}} \sum_{i=1}^{\infty} \left(\frac{\lambda_i}{\lambda + \lambda_i}\right)^2 e_i(x)^2 \mathrm{d}\mu(x)$$

$$= \sum_{i=1}^{\infty} \left(\frac{\lambda_i}{\lambda + \lambda_i}\right)^2 =: \mathcal{N}_2(\lambda).$$

It can be shown that $\mathcal{N}_2(\lambda) = \mathrm{Tr}\left[(T + \lambda)^{-1} T\right]^2$ and it is a variant of the effective dimension $\mathcal{N}(\lambda) = \mathrm{Tr}\left[(T + \lambda)^{-1} T\right]$ introduced in the literature (Caponnetto & De Vito, 2007). The condition (**A**) implies that $\mathcal{N}_2(\lambda) \geq c\lambda^{-\beta}$. As a result, we get

$$\textbf{Var} \geq \frac{\bar{\sigma}^2}{n} \int_{\mathcal{X}} \left\|(T_X + \lambda)^{-1} k(x, \cdot)\right\|_{L^2, n}^2 \mathrm{d}\mu(x) \approx \frac{\bar{\sigma}^2}{n} \int_{\mathcal{X}} \left\|(T + \lambda)^{-1} k(x, \cdot)\right\|_{L^2}^2 \mathrm{d}\mu(x)$$

$$= \frac{\bar{\sigma}^2}{n} \mathcal{N}_2(\lambda) \geq \tilde{c} \frac{\lambda^{-\beta}}{n}.$$

## 4 Numerical Experiments

The saturation effect in KRR has been reported in a number of works (e.g., Gerfo et al. (2008); Dicker et al. (2017)). In this section, we illustrate the saturation effect through a toy example.

Suppose that $\mathcal{X} = [0, 1]$ and $\mu$ is the uniform distribution on $[0, 1]$. Let us consider the following first-order Sobolev space containing absolutely continuous functions

$$H^1 := \left\{ f : [0, 1] \to \mathbb{R} \,\Big|\, f \text{ is A.C.}, f(0) = 0, \int_0^1 (f'(x))^2 \mathrm{d}x < \infty \right\}. \tag{24}$$

It is well-known that it is the RKHS associated to the kernel $k(x, y) = \min(x, y)$ (Wainwright, 2019). Let $T$ be the integral operator associated to the kernel function $k$. We know explicitly the eigenvalues and eigenfunctions of this operator:

$$\lambda_n = \left( \frac{2n-1}{2}\pi \right)^{-2}, \quad e_n(x) = \sqrt{2} \sin\left( \frac{2n-1}{2}\pi x \right), \quad n = 1, 2, \ldots. \tag{25}$$

It is clear that the eigenvalue decay rate $\beta$ of the kernel function $k$ (or the RKHS $H^1$) is $0.5$. To illustrate the saturation effect better, we choose the second eigen-function $e_2 = \sqrt{2}\sin\left(\frac{3}{2}\pi x\right)$ to be the regression function in our experiment. The significance of this choice is that for any $\alpha > 0$, we have $e_2 \in [\mathcal{H}]^\alpha$. We further set the noise to be an independent Gaussian noise with variance $\sigma = 0.2$. In other words, we consider the following data generation model:

$$y = f^*(x) + \sigma\varepsilon, \tag{26}$$

where $f^*(x) = e_2(x)$ and $\varepsilon \sim N(0, 1)$ is the standard normal distribution.

Since the gradient flow (GF) method with proper early stopping is a spectral algorithm proved to be rate optimal for any $\alpha \geq 1$ and any $f_\rho^* \in [\mathcal{H}]^\alpha$ (Yao et al., 2007; Lin et al., 2018), we make a comparison between KRR and GF to show the saturation effect. More precisely, we report and compare the decaying rates of the generalization errors produced by the KRR and GF methods with different selections of parameters.

For various $\alpha$'s, we choose the regularization parameter in KRR as $\lambda = cn^{-\frac{1}{\alpha+\beta}}$ for a fixed constant $c$, and set the stopping time in the gradient flow by $t = \lambda^{-1}$. It is shown in Lin et al. (2018) that the choice of stopping time $t$ (i.e., $t = \lambda^{-1}$) is optimal for the gradient descent algorithm under the assumption that $f_\rho^* \in [\mathcal{H}]^\alpha$. By choosing different $\alpha$'s, we also evaluate the performance of the algorithms with different selections of parameters. For the generalization error $\|\hat{f} - f_\rho^*\|_{L^2}^2$, we numerically compute the integration ($L^2$-norm) by Simpson's formula with $N \gg n$ points. For each $n$, we perform 100 trials and show the average as well as the region within one standard deviation. Finally, we use logarithmic least-squares $\log \mathrm{err} = r \log n + b$ to fit the error with respect to the sample size, and report the slope $r$ as the convergence rate.

The results are reported in Figure 1 on page 9. We also change the regression function to be other eigenfunctions and list the results in Table 4 on page 9. First, the error curves show that the error converges indeed in the rate of $n^{-r}$. Moreover, when we apply the GF method, the convergence rate increases as the $\alpha$ increases, confirming that spectral algorithms with high qualification can adapt to the smoothness of the regression function. The convergence rates also match the theoretical value $\frac{\alpha}{\alpha+\beta}$. In contrast, when we apply the KRR method, the convergence rate achieves its best performance at $\alpha = 2$, and the rate decreases as $\alpha$ gets bigger, showing the saturation effect. The resulting best rate also coincides with our theoretic value $\frac{2}{2+\beta} = 0.8$. We also remark that besides the best rate, rates of other selection of regularization parameter also correspond to theoretical lower bounds that can be further obtained by the bias-variance decomposition (22).

We conduct further experiments with different kernels and report them in Section E. The results are also approving.

In conclusion, our numerical results confirm the saturation effect and approve our theory, and all the results can be explained and understood by the theory.

## 5 CONCLUSION

The saturation effect refers to the phenomenon that kernel ridge regression fails to achieve the information theoretical lower bound when the regression function is too smooth. When the regression function is sufficiently smooth, a saturation lower bound of KRR has been conjectured for decades.

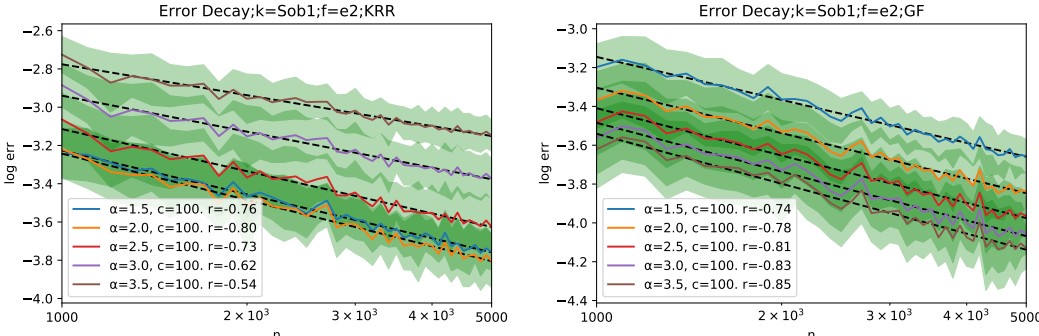

Figure 1: Error decay curves of KRR and GF. Both axes are logarithmic. The colored curves show the averaged error over 100 trials and the regions within one standard deviation are shown in green. The dashed black lines are computed using logarithmic least-squares and the slopes are reported as convergence rates.

| $\alpha$ | $f^* = e_1$ | | $f^* = e_2$ | | $f^* = e_3$ | | $f^* = e_4$ | |
|---|---|---|---|---|---|---|---|---|
| | KRR | GF | KRR | GF | KRR | GF | KRR | GF |
| 1.5 | .73 | .72 | .76 | .74 | .75 | .74 | .74 | .75 |
| 2.0 | **.76** | .76 | **.80** | .78 | **.78** | .78 | **.80** | .80 |
| 2.5 | .69 | .80 | .73 | .81 | .69 | .82 | .67 | .84 |
| 3.0 | .58 | .82 | .62 | .83 | .59 | .84 | .56 | .87 |
| 3.5 | .49 | **.84** | .54 | **.85** | .51 | **.86** | .48 | **.90** |

Table 1: Convergence rates comparison between KRR and GF with $\lambda = cn^{-\frac{1}{\alpha+\beta}}$ for various $\alpha$'s. Bold numbers represent the max rate over different choices of $\lambda$.

In this paper, we provide a rigorously proof of the saturation effect of KRR, i.e., we show that, if $f \in [\mathcal{H}]^\alpha$ for some $\alpha > 2$, the rate of generalization error of the KRR can not be better than $n^{-\frac{2}{2+\beta}}$, no matter how one tunes the KRR.

Our results suggest that the KRR method of regularization may be inferior to some special regularization algorithms, including spectral cut-off and kernel gradient descent, which never saturate and are capable of achieving optimal rates (Bauer et al., 2007; Lin et al., 2018). The technical tools developed here may also help us establish the lower bound of the saturation effects for other spectral regularization algorithms.

### ACKNOWLEDGMENTS

This research was partially supported by the National Natural Science Foundation of China (Grant 11971257), Beijing Natural Science Foundation (Grant Z190001), National Key R&D Program of China (2020AAA0105200), and Beijing Academy of Artificial Intelligence.

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

# A    BASIC FACTS IN RKHS

Let $k$ be a continuous positive definite kernel function defined on a compact set $\mathcal{X}$ and $\mathcal{H}$ be the RKHS associated to the kernel $k$. Since $k$ is a continuous function and $\mathcal{X}$ is a compact set, there exists a constant $\kappa$ such that

$$|k(x, y)| \leq \kappa \text{ for any } x, y \in \mathcal{X}. \tag{27}$$

It is well know that $k(x, \cdot) \in \mathcal{H}$ as a function and that the inner product satisfies that

$$\langle k(x, \cdot), f \rangle = f(x), \quad \forall f \in \mathcal{H}. \tag{28}$$

In particularly, we have $\langle k(x, \cdot), k(y, \cdot) \rangle_{\mathcal{H}} = k(x, y)$.

Let $T : L^2(\mathcal{X}) \to L^2(\mathcal{X})$ be the integral operator associated to the kernel fucntion $k$. By the spectral decomposition of $T$, it can be easily shown that $\operatorname{Ran} T|_{\mathcal{H}} \subseteq \mathcal{H}$ and $\|Tf\|_{\mathcal{H}} \leq \kappa^2 \|f\|_{\mathcal{H}}$, which implies that $T$ can also be viewed a bounded linear operator on $\mathcal{H}$. With a little abuse of notation, we still use $T$ to indicate it.

We denote by $\mathscr{B}(\mathcal{H})$ the set of bounded linear operators over $\mathcal{H}$ and $\|\cdot\|_{\mathscr{B}(\mathcal{H})}$ the corresponding operator norm, where the subscript may be omitted if there is no confusion.

## A.1    FUNCTIONS IN RKHS

**Lemma A.1.** *Suppose that $\mathcal{H}$ is the RKHS associated to the kernel $k$. Let $\|f\|_{\infty} := \sup_{x \in \mathcal{X}} |f(x)|$ be the supremum-norm of a function $f : \mathcal{X} \to \mathbb{R}$. Then for any $f \in \mathcal{H}$, we have $\|f\|_{\infty} \leq \kappa \|f\|_{\mathcal{H}}$*

*Proof.* It is easy to verify that for $f \in \mathcal{H}$,

$$\begin{aligned}
\|f\|_{\infty} = \sup_{x \in \mathcal{X}} |f(x)| &\leq \sup_{x \in \mathcal{X}} |\langle k(x, \cdot), f \rangle_{\mathcal{H}}| \\
&\leq \sup_{x \in \mathcal{X}} \|k(x, \cdot)\|_{\mathcal{H}} \|f\|_{\mathcal{H}} \leq \kappa \|f\|_{\mathcal{H}}.
\end{aligned}$$

$\square$

**Definition A.2.** *Let $K \subseteq \mathbb{R}^d$ be a compact set and $\alpha \in [0, 1]$. For a function $f : K \to \mathbb{R}$, we introduce the Hölder semi-norm*

$$[f]_{\alpha, K} := \sup_{x, y \in K, \ x \neq y} \frac{|f(x) - f(y)|}{|x - y|^{\alpha}}, \tag{29}$$

*where $|\cdot|$ represents the usual Euclidean norm. Then, we define the Hölder space*

$$C^{\alpha}(K) := \{f : K \to \mathbb{R} \mid [f]_{\alpha, K} < \infty\}, \tag{30}$$

*which is equipped with norm*

$$\|f\|_{C^{\alpha}(K)} := \sup_{x \in K} |f(x)| + [f]_{\alpha, K}.$$

**Lemma A.3.** *Assume that $\mathcal{H}$ is an RKHS over a compact set $\mathcal{X} \subseteq \mathbb{R}^d$ associated with a kernel $k \in C^{\alpha}(\mathcal{X} \times \mathcal{X})$ for $\alpha \in (0, 1]$. Then, we have $\mathcal{H} \subseteq C^{\alpha/2}(\mathcal{X})$ and*

$$[f]_{\alpha/2, \mathcal{X}} \leq \sqrt{2\kappa^2 [k]_{\alpha, \mathcal{X} \times \mathcal{X}}} \|f\|_{\mathcal{H}}, \tag{31}$$

*where $\kappa^2 := \sup_{x \in \mathcal{X}} |k(x, x)|$.*

*Proof.* Since $k$ is positive definite, from $\det \begin{vmatrix} k(x, x) & k(x, y) \\ k(x, y) & k(y, y) \end{vmatrix} \geq 0$ we know that $\sup_{x, y \in \mathcal{X}} |k(x, y)| \leq \kappa^2$. By the properties of RKHS, we have

$$f(x) - f(y) = \langle f, k(x, \cdot) \rangle_{\mathcal{H}} - \langle f, k(y, \cdot) \rangle_{\mathcal{H}} = \langle f, k(x, \cdot) - k(b, \cdot) \rangle_{\mathcal{H}} \leq \|f\|_{\mathcal{H}} \|k(x, \cdot) - k(y, \cdot)\|_{\mathcal{H}}.$$

Moreover,

$$\|k(x,\cdot) - k(y,\cdot)\|_{\mathcal{H}}^2 = k(x,x)k(y,y) - k(x,y)^2$$
$$\leq k(x,x)|k(y,y) - k(x,y)| + k(x,y)|k(x,x) - k(x,y)| \leq 2\kappa^2[k]_{\alpha,\mathcal{X}\times\mathcal{X}}|x-y|^\alpha.$$

Therefore, we obtain

$$|f(x) - f(y)| \leq \|f\|_{\mathcal{H}}\sqrt{2\kappa^2[k]_{\alpha,\mathcal{X}\times\mathcal{X}}}|x-y|^{\alpha/2}.$$

$\square$

## A.2 SAMPLE SUBSPACE AND SEMI-NORM

It is convenient to introduce the following commonly used notations

$$\mathbb{K}(x,X) \coloneqq (k(x,x_1),\ldots,k(x,x_n)), \quad \mathbb{K}(X,x) \coloneqq \mathbb{K}(x,X)', \tag{32}$$

$$\mathbb{K}(X,X) \coloneqq \big(k(x_i,x_j)\big)_{i,j}, \tag{33}$$

$$K \coloneqq \frac{1}{n}\mathbb{K}(X,X), \tag{34}$$

where $K$ is known as the (normalized) kernel matrix. For a function $f$, we also denote by

$$f[X] = (f(x_1),\ldots,f(x_n))'$$

the column vector of function values.

**Definition A.4.** *Given $\{x_1,...,x_n\} \subset \mathcal{X}$, the subspace*

$$\mathcal{H}_n \coloneqq \operatorname{span}\{k(x_1,\cdot),\ldots,k(x_n,\cdot)\} \subset \mathcal{H}. \tag{35}$$

*of $\mathcal{H}$ is called the sample subspace. We also call the operator $Q_n : \mathcal{H} \to \mathcal{H}_n$ given by*

$$Q_n(f)(x) = \mathbb{K}(x,X)\mathbb{K}(X,X)^{-1}f(X). \tag{36}$$

*the sample projection map.*

Recall that we have defined the operator $T_X = \frac{1}{n}\sum_i K_{x_i}K_{x_i}^*$ in (19). It is clear that $T_X = T_X Q_n$ and $\operatorname{Ran} T_X = \mathcal{H}_n$. Under the natural base $\{k(x_1,\cdot),\ldots,k(x_1,\cdot)\}$ of $\mathcal{H}_n$, we have

$$T_X k(x_i,\cdot) = \frac{1}{n}\sum_{j=1}^n K_{x_j}k(x_i,x_j) = \frac{1}{n}k(x_i,x_j)k(x_j,\cdot),$$

i.e., $T_X$ can be represented by the matrix $K$ under the natural basis. We can use

$$T_X\mathbb{K}(X,\cdot) = K\mathbb{K}(X,\cdot) \tag{37}$$

to express this result. Furthermore, for any continuous function $\varphi(x)$, the operator $\varphi(T_X)$ satisfies that

$$\varphi(T_X)\mathbb{K}(X,\cdot) = \varphi(K)\mathbb{K}(X,\cdot). \tag{38}$$

In particlar, we have

$$(\varphi(T_X)f)[X] = \varphi(K)f[X]. \tag{39}$$

Since $g_Z \in \mathcal{H}_n$ ( see (20)). Therefore, we know that

$$\hat{f}_\lambda^{\mathrm{KRR}}(x) = \frac{1}{n}\mathbb{K}(x,X)\left(K+\lambda\right)^{-1}\mathbf{y}.$$

### A.2.1 Semi-inner products in the sample space

We consider the following sample semi-inner products:

$$\langle f, g \rangle_{L^2,n} := \frac{1}{n} \sum_{i=1}^{n} f(x_i) g(x_i) = \frac{1}{n} f[X]' g[X], \tag{40}$$

$$\langle f, g \rangle_{\mathcal{H},n} := \langle Q_n f, Q_n g \rangle_{\mathcal{H}}. \tag{41}$$

**Lemma A.5.**

$$\langle f, g \rangle_{\mathcal{H},n} = \frac{1}{n} f[X]' K^{-1} g[X] \tag{42}$$

*Proof.* By the definition (36) of $Q_n$, we have

$$
\begin{aligned}
\langle f, g \rangle_{\mathcal{H},n} &= \left\langle \mathbb{K}(\cdot, X) \mathbb{K}(X, X)^{-1} f[X], \mathbb{K}(\cdot, X) \mathbb{K}(X, X)^{-1} g[X] \right\rangle_{\mathcal{H}} \\
&= \left( \mathbb{K}(X, X)^{-1} f[X] \right)' \mathbb{K}(X, X) \left( \mathbb{K}(X, X)^{-1} g[X] \right) \\
&= f[X]' \mathbb{K}(X, X)^{-1} g[X] = \frac{1}{n} f[X]' K^{-1} g[X].
\end{aligned}
$$

$\square$

**Proposition A.6.** *For $f, g \in \mathcal{H}$, we have*

$$
\begin{aligned}
\langle f, g \rangle_{L^2,n} &= \langle T_X f, g \rangle_{\mathcal{H}} = \left\langle T_X^{1/2} f, T_X^{1/2} g \right\rangle, \\
\| f \|_{L^2,n} &= \left\| T_X^{1/2} f \right\|_{\mathcal{H}}.
\end{aligned}
\tag{43}
$$

*Proof.* Since $\operatorname{Ran} T_X = \mathcal{H}_n$, we have

$$\langle T_X f, g \rangle_{\mathcal{H}} = \langle Q_n T_X f, Q_n g \rangle = \langle T_X f, g \rangle_{\mathcal{H},n}.$$

Since $(T_X f)[X] = K f[X]$, we obtain

$$\langle T_X f, g \rangle_{\mathcal{H},n} = \frac{1}{n} [(T_X f)[X]]' K^{-1} g[X] = \frac{1}{n} f[X]' g[X] = \langle f, g \rangle_{L^2,n}.$$

$\square$

### A.3 Covering number and entropy number

**Definition A.7.** *Let $(E, \|\cdot\|_E)$ be a normed space and $A \subset E$ be a subset. For $\varepsilon > 0$, we say $S \subseteq A$ is an $\varepsilon$-net of $A$ if $\forall a \in A$, $\exists s \in S$ such that $\|a - s\| \leq \varepsilon$. Moreover, we define the $\varepsilon$-covering number of $A$ to be*

$$\mathcal{N}(A, \|\cdot\|_E, \varepsilon) := \inf \left\{ n \in \mathbb{N}^* : \exists s_1, \ldots, s_n \in A \text{ such that } A \subseteq \bigcup_{i=1}^{n} B_E(s_i, \varepsilon) \right\}, \tag{44}$$

$$= \inf \left\{ |S| : S \text{ is an } \varepsilon\text{-net of } A \right\} \tag{45}$$

*where $B_E(x_0, \varepsilon) := \{ x \in E \mid \|x_0 - x\|_E \leq \varepsilon \}$ be the closed ball centered at $x_0 \in E$ with radius $\varepsilon$.*

The following result about the covering number of a bounded set in the Euclidean space is well-known, see, e.g., Vershynin (2018, Section 4.2).

**Lemma A.8.** *Let $A \subseteq \mathbb{R}^d$ be a bounded set. Then there exists a constant $C$ (depending on $A$) such that*

$$\mathcal{N}\left( A, \|\cdot\|_{\mathbb{R}^d}, \varepsilon \right) \leq C \varepsilon^{-d}. \tag{46}$$

# B  PROOF OF THE MAIN THEOREM

## B.1  BIAS-VARIANCE DECOMPOSITION

The first step of the proof is the traditional bias-variance decomposition. Recalling (21), we have

$$\hat{f}_\lambda^{\text{KRR}} = \frac{1}{n}(T_X + \lambda)^{-1} \sum_{i=1}^n K_{x_i} y_i = \frac{1}{n}(T_X + \lambda)^{-1} \sum_{i=1}^n K_{x_i}(K_{x_i}^* f^* + \epsilon_i)$$

$$= (T_X + \lambda)^{-1} T_X f^* + \frac{1}{n} \sum_{i=1}^n (T_X + \lambda)^{-1} K_{x_i} \epsilon_i,$$

so that

$$\hat{f}_\lambda^{\text{KRR}} - f^* = -\lambda(T_X + \lambda)^{-1} f^* + \frac{1}{n} \sum_{i=1}^n (T_X + \lambda)^{-1} K_{x_i} \epsilon_i.$$

Taking expectation over the noise $\epsilon$ conditioned on $X$, since $\varepsilon|x$ are independent noise with mean 0 and variance $\sigma_x^2$, we have

$$\mathbb{E}\left(\left\|\hat{f}_\lambda^{\text{KRR}} - f^*\right\|_{L^2}^2 \;\Big|\; X\right) = \mathbf{Bias}^2 + \mathbf{Var}, \tag{47}$$

where

$$\mathbf{Bias}^2 := \lambda^2 \left\|(T_X + \lambda)^{-1} f^*\right\|_{L^2}^2, \quad \mathbf{Var} := \frac{1}{n^2} \sum_{i=1}^n \sigma_{x_i}^2 \left\|(T_X + \lambda)^{-1} k(x_i, \cdot)\right\|_{L^2}^2. \tag{48}$$

## B.2  LOWER BOUND FOR THE BIAS TERM

**Proposition B.1.** *Suppose that $f^* \in [\mathcal{H}]^0$ is a non-zero function. There is some $c > 0$ independent of $\lambda$ such that*

$$\left\|(T + \lambda)^{-1} f^*\right\|_{L^2}^2 \geq c \quad as \quad \lambda \to 0. \tag{49}$$

*Proof.* Since $f^* \in [\mathcal{H}]^0$ and it is non-zero, we may assume that

$$f^* = \sum_{i=1}^\infty a_i e_i.$$

Because $\lambda \to 0$, we have

$$\left\|(T + \lambda)^{-1} f^*\right\|_{L^2}^2 = \sum_{i=1}^\infty \left(\frac{a_i}{\lambda_i + \lambda}\right)^2 \geq \sum_{i=1}^\infty \left(\frac{a_i}{\lambda_i + 1}\right)^2 > 0 \quad (\text{Since } f^* \neq 0).$$

$\square$

**Theorem B.2** (Lower bound of the Bias term). *Suppose that $\alpha \geq 2$, $f^* \in [\mathcal{H}]^\alpha$ is a non-zero function and $\lambda = \lambda(n) = \Omega(n^{-(1-\varepsilon)})$ for some $\varepsilon \in (0, 1)$. Then, for any $\delta > 0$, there exists an integer $n_0$ such that for any $n > n_0$, we have that*

$$\mathbf{Bias}^2 \geq c\lambda^2, \tag{50}$$

*holds with probability at least $1 - \delta$ where $c > 0$ is a constant independent of $\lambda$. As a consequence, $\mathbf{Bias}^2 = \Omega_{\mathbb{P}}(\lambda^2)$.*

*Proof.* Corollary C.8 together with Proposition B.1 yields that with probability at least $1 - \delta$ we have

$$\left\|(T_X + \lambda)^{-1} f^*\right\|_{L^2}^2 \geq \left\|(T + \lambda)^{-1} f^*\right\|_{L^2}^2 - O(n^{-q}) \left(\ln \frac{4}{\delta}\right)^2$$

$$\geq c - O(n^{-q}) \left( \ln \frac{4}{\delta} \right)^2$$

for some $q > 0$. Therefore, we get

$$\mathbf{Bias}^2 = \lambda^2 \left\| (T_X + \lambda)^{-1} f^* \right\|_{L^2}^2 \geq c\lambda^2 \left( 1 - O(n^{-q}) \left( \ln \frac{4}{\delta} \right)^2 \right) \geq \frac{c}{2} \lambda^2$$

when $n$ is sufficiently large. $\qquad\qquad\square$

### B.3 LOWER BOUND FOR THE VARIANCE TERM

For the variance term, Assumption 2 yields that $\sigma_{x_i}^2 \geq \bar{\sigma}^2$ almost surely. Recalling the discussion of sample subspaces in Section A.2, we have

$$
\begin{aligned}
\mathbf{Var} &\geq \frac{\bar{\sigma}^2}{n^2} \sum_{i=1}^n \left\| (T_X + \lambda)^{-1} k(x_i, \cdot) \right\|_{L^2}^2 \\
&= \frac{\bar{\sigma}^2}{n^2} \left\| (T_X + \lambda)^{-1} \mathbb{K}(X, \cdot) \right\|_{L^2(\mathcal{X}, \mathrm{d}\mu; \mathbb{R}^n)}^2 \\
(\text{By (38)}) \quad &= \frac{\bar{\sigma}^2}{n^2} \left\| (K + \lambda)^{-1} \mathbb{K}(X, \cdot) \right\|_{L^2(\mathcal{X}, \mathrm{d}\mu; \mathbb{R}^n)}^2 \\
&= \frac{\bar{\sigma}^2}{n^2} \int_{\mathcal{X}} \mathbb{K}(x, X)(K + \lambda)^{-2} \mathbb{K}(X, x) \mathrm{d}\mu(x).
\end{aligned}
\tag{51}
$$

Let us denote $h_x = k(x, \cdot)$. Then by definition it is obvious that $h_x[X] = \mathbb{K}(X, x)$. From (39), we find that

$$\left( (T_X + \lambda)^{-1} h_x \right)[X] = (K + \lambda)^{-1} h_x[X] = (K + \lambda)^{-1} \mathbb{K}(X, x),$$

so we obtain

$$
\begin{aligned}
\frac{1}{n} \mathbb{K}(x, X)(K + \lambda)^{-2} \mathbb{K}(X, x) &= \frac{1}{n} \left\| (K + \lambda)^{-1} \mathbb{K}(X, x) \right\|_{\mathbb{R}^n}^2 \\
&= \frac{1}{n} \left\| \left( (T_X + \lambda)^{-1} h_x \right)[X] \right\|_{\mathbb{R}^n}^2 \\
&= \left\| (T_X + \lambda)^{-1} h_x \right\|_{L^2, n}^2.
\end{aligned}
$$

from the definition (40) of sample semi-inner product. Consequently, we get

$$\mathbf{Var} \geq \frac{\bar{\sigma}^2}{n} \int_{\mathcal{X}} \left\| (T_X + \lambda)^{-1} h_x \right\|_{L^2, n}^2 \mathrm{d}\mu(x). \tag{52}$$

Combining with some concentration results, we can obtain the following theorem.

**Theorem B.3.** *Assume that Assumptions 1 and 2 and condition* (**A**) *(e.g., the eigenvalue decay rate* (9)*) hold. Suppose that* $\lambda = \lambda(n) \to 0$ *satisfying that* $\lambda = \Omega\left( n^{-\frac{1}{2}+p} \right)$ *for some* $p \in (0, 1/2)$. *Then, for any* $\delta > 0$, *when* $n$ *is sufficiently large, the following holds with probability at least* $1 - \delta$:

$$\mathbf{Var} \geq \frac{c\lambda^{-\beta}}{n}. \tag{53}$$

*As a consequence, we have* $\mathbf{Var} = \Omega_{\mathbb{P}} \left( \frac{\lambda^{-\beta}}{n} \right)$.

*Proof.* First, we assert that the approximation

$$\left\| (T_X + \lambda)^{-1} h_x \right\|_{L^2, n}^2 \geq \frac{1}{2} \left\| (T + \lambda)^{-1} h_x \right\|_{L^2}^2 - \left( o(1) \left\| (T + \lambda)^{-1} h_x \right\|_{L^2} + o(1) \ln \frac{4}{\delta} \right) \ln \frac{4}{\delta} \tag{54}$$

holds with probability at least $1 - \delta$. Then, plugging the approximation into (52) gives

$$
\begin{aligned}
\mathbf{Var} &\geq \frac{\bar{\sigma}^2}{n} \int_{\mathcal{X}} \left\| (T_X + \lambda)^{-1} h_x \right\|_{L^2, n}^2 \mathrm{d}\mu(x) \\
&\geq \frac{\bar{\sigma}^2}{2n} \int_{\mathcal{X}} \left\| (T + \lambda)^{-1} h_x \right\|_{L^2}^2 \mathrm{d}\mu(x) - \frac{o(1)}{n} \int_{\mathcal{X}} \left\| (T + \lambda)^{-1} h_x \right\|_{L^2} \mathrm{d}\mu(x) - \frac{o(1)}{n} \left( \ln \frac{4}{\delta} \right)^2 .
\end{aligned}
$$

For the two integral terms, applying Mercer's theorem, we get

$$
\begin{aligned}
\int_{\mathcal{X}} \left\| (T + \lambda)^{-1} h_x \right\|_{L^2}^2 \mathrm{d}\mu(x) &= \int_{\mathcal{X}} \sum_{i=1}^{\infty} \left( \frac{\lambda_i}{\lambda + \lambda_i} \right)^2 e_i(x)^2 \mathrm{d}\mu(x) \\
&= \sum_{i=1}^{\infty} \left( \frac{\lambda_i}{\lambda + \lambda_i} \right)^2 = \mathcal{N}_2(\lambda) \geq c\lambda^{-\beta},
\end{aligned}
\tag{55}
$$

and

$$
\begin{aligned}
\int_{\mathcal{X}} \left\| (T + \lambda)^{-1} h_x \right\|_{L^2} \mathrm{d}\mu(x) &\leq \left( \int_{\mathcal{X}} \left\| (T + \lambda)^{-1} h_x \right\|_{L^2}^2 \mathrm{d}\mu(x) \right)^{1/2} \\
&= (\mathcal{N}_2(\lambda))^{1/2} \leq C\lambda^{-\beta/2},
\end{aligned}
$$

where the estimation of $\mathcal{N}_2(\lambda)$ comes from Proposition D.1. Therefore, we obtain that

$$
\mathbf{Var} \geq \frac{c\bar{\sigma}^2}{2n} \lambda^{-\beta} - \frac{o(\lambda^{-\beta/2})}{n} \ln \frac{4}{\delta} - \frac{o(1)}{n} \left( \ln \frac{4}{\delta} \right)^2 \geq \frac{c\bar{\sigma}^2}{4n} \lambda^{-\beta}
$$

as $n$ goes to infinity.

It remains to establish the approximation (54). Lemma C.11 and Lemma C.12 yield that

$$
\left\| (T + \lambda)^{-1} h_x \right\|_{L^2, n}^2 \leq \frac{3}{2} \left\| (T + \lambda)^{-1} h_x \right\|_{L^2}^2 + o(1) \ln \frac{4}{\delta},
\tag{56}
$$

$$
\left\| (T + \lambda)^{-1} h_x \right\|_{L^2, n}^2 \geq \frac{1}{2} \left\| (T + \lambda)^{-1} h_x \right\|_{L^2}^2 - o(1) \ln \frac{4}{\delta},
\tag{57}
$$

$$
\left| \left\| T_X^{1/2} (T_X + \lambda)^{-1} h_x \right\|_{\mathcal{H}} - \left\| T_X^{1/2} (T + \lambda)^{-1} h_x \right\|_{\mathcal{H}} \right| \leq o(1) \ln \frac{4}{\delta}
\tag{58}
$$

with probability at least $1 - \delta$. Consequently, from (43) and (56), we get

$$
\begin{aligned}
\left\| T_X^{1/2} (T + \lambda)^{-1} h_x \right\|_{\mathcal{H}} &= \left\| (T + \lambda)^{-1} h_x \right\|_{L^2, n} \\
&\leq C \left\| (T + \lambda)^{-1} h_x \right\|_{L^2} + o(1) \left( \ln \frac{4}{\delta} \right)^{1/2} .
\end{aligned}
$$

Combining it with (58), we find that

$$
\begin{aligned}
\left\| T_X^{1/2} (T_X + \lambda)^{-1} h_x \right\|_{\mathcal{H}} + \left\| T_X^{1/2} (T + \lambda)^{-1} h_x \right\|_{\mathcal{H}} &\leq 2 \left\| T_X^{1/2} (T + \lambda)^{-1} h_x \right\|_{\mathcal{H}} + o(1) \ln \frac{4}{\delta} \\
&\leq C \left\| (T + \lambda)^{-1} h_x \right\|_{L^2} + o(1) \ln \frac{4}{\delta},
\end{aligned}
$$

which gives the approximation of the squared norm

$$
\begin{aligned}
\left| \left\| T_X^{1/2} (T_X + \lambda)^{-1} h_x \right\|_{\mathcal{H}}^2 - \left\| T_X^{1/2} (T + \lambda)^{-1} h_x \right\|_{\mathcal{H}}^2 \right| \\
\leq o(1) \ln \frac{4}{\delta} \cdot \left( \left\| (T + \lambda)^{-1} h_x \right\|_{L^2} + o(1) \ln \frac{4}{\delta} \right).
\end{aligned}
\tag{59}
$$

Finally, combining (59) and (57) yields

$$
\left\| (T_X + \lambda)^{-1} h_x \right\|_{L^2, n}^2 = \left\| T_X^{1/2} (T_X + \lambda)^{-1} h_x \right\|_{\mathcal{H}}^2
$$

$$
\begin{aligned}
&\geq \left\| T_X^{1/2}(T+\lambda)^{-1}h_x \right\|_{\mathcal{H}}^2 - o(1)\ln\frac{4}{\delta} \cdot \left( \left\|(T+\lambda)^{-1}h_x\right\|_{L^2} + o(1)\ln\frac{4}{\delta} \right) \\
&= \left\|(T+\lambda)^{-1}h_x\right\|_{L^2,n}^2 - o(1)\ln\frac{4}{\delta} \cdot \left( \left\|(T+\lambda)^{-1}h_x\right\|_{L^2} + o(1)\ln\frac{4}{\delta} \right) \\
&\geq \frac{1}{2}\left\|(T+\lambda)^{-1}h_x\right\|_{L^2}^2 - o(1)\ln\frac{4}{\delta} - o(1)\ln\frac{4}{\delta} \cdot \left( \left\|(T+\lambda)^{-1}h_x\right\|_{L^2} + o(1)\ln\frac{4}{\delta} \right) \\
&= \frac{1}{2}\left\|(T+\lambda)^{-1}h_x\right\|_{L^2}^2 - \left( o(1)\left\|(T+\lambda)^{-1}h_x\right\|_{L^2} + o(1)\ln\frac{4}{\delta} \right)\ln\frac{4}{\delta}.
\end{aligned}
$$

$\square$

### B.4 Proof of Theorem 3.1

Let $\lambda = \lambda(n)$ be an arbitrary choice of regularization parameter satisfying that $\lambda(n) \to 0$. We consider the truncation

$$
\bar{\lambda} := \max\left( \lambda, n^{-\frac{1}{2+\beta}} \right), \tag{60}
$$

which satisfies that $\bar{\lambda} = \Omega\left( n^{-\frac{1}{2+\beta}} \right)$ and $\bar{\lambda} \to 0$. Applying Theorem B.2 and Theorem B.3 to $\bar{\lambda}$, we obtain that

$$
\mathbf{Bias}^2(\bar{\lambda}) \geq c_1\bar{\lambda}^2, \qquad \mathbf{Var}(\bar{\lambda}) \geq \frac{c_2\bar{\lambda}^{-\beta}}{n}
$$

with probability at least $1 - \delta$ for sufficiently large $n$, where we use $\mathbf{Bias}^2(\bar{\lambda})$ and $\mathbf{Var}(\bar{\lambda})$ to highlight the choice of regularization parameter. Let us consider two cases.

**Case 1:** $\lambda > n^{-\frac{1}{2+\beta}}$   In this case $\lambda = \bar{\lambda}$, so

$$
\begin{aligned}
\mathbb{E}\left( \left\| \hat{f}_\lambda^{\mathrm{KRR}} - f^* \right\|^2 \,\Big|\, X \right) &= \mathbf{Bias}^2(\lambda) + \mathbf{Var}(\lambda) \\
&= \mathbf{Bias}^2(\bar{\lambda}) + \mathbf{Var}(\bar{\lambda}) \\
&\geq c_1\bar{\lambda}^2 + \frac{c_2\bar{\lambda}^{-\beta}}{n} \\
&\geq cn^{-\frac{2}{2+\beta}},
\end{aligned}
$$

where the last inequality is obtained by elementary inequalities in Lemma D.3 with $\frac{1}{p} = \frac{\beta}{2+\beta}$ and $\frac{1}{q} = \frac{2}{2+\beta}$.

**Case 2:** $\lambda \leq n^{-\frac{1}{2+\beta}}$   From the intermediate result (51) in proving the lower bound of the variance, we know that

$$
\mathbf{Var}(\bar{\lambda}) \geq \frac{\bar{\sigma}^2}{n^2}\int_{\mathcal{X}} \mathbb{K}(x,X)(K+\bar{\lambda})^{-2}\mathbb{K}(X,x)\mathrm{d}\mu(x) \geq \frac{c_2\bar{\lambda}^{-\beta}}{n}
$$

and

$$
\mathbf{Var}(\lambda) \geq \frac{\bar{\sigma}^2}{n^2}\int_{\mathcal{X}} \mathbb{K}(x,X)(K+\lambda)^{-2}\mathbb{K}(X,x)\mathrm{d}\mu(x).
$$

Noticing that

$$
(K+\lambda_1)^{-2} \succeq (K+\lambda_2)^{-2} \quad\text{if}\quad \lambda_1 \leq \lambda_2,
$$

where $\succeq$ represents the partial order induced by positive definite matrices, we get

$$
\mathbf{Var}(\lambda) \geq \frac{\bar{\sigma}^2}{n^2}\int_{\mathcal{X}} \mathbb{K}(x,X)(K+\lambda)^{-2}\mathbb{K}(X,x)\mathrm{d}\mu(x)
$$

$$\geq \frac{\bar{\sigma}^2}{n^2} \int_{\mathcal{X}} \mathbb{K}(x, X)(K + \bar{\lambda})^{-2} \mathbb{K}(X, x) \mathrm{d}\mu(x)$$

$$\geq \frac{c_2 \bar{\lambda}^{-\beta}}{n} = c_2 n^{-\frac{2}{2+\beta}},$$

where we note that $\bar{\lambda} = n^{-\frac{1}{2+\beta}}$ in this case. Consequently,

$$\mathbb{E}\left(\left\|\hat{f}_\lambda^{\mathrm{KRR}} - f^*\right\|^2 \mid X\right) \geq \mathbf{Var}(\lambda) \geq c_2 n^{-\frac{2}{2+\beta}}.$$

The proof is completed by concluding two cases.

**Remark B.4.** It is worth noticing that both the requirement that $f^* \neq 0$ and that the noise is non-vanishing are necessary. If the former does not hold, choosing $\lambda = \infty$ will yield the best estimator $\hat{f} = 0$ with zero loss. If the latter does not hold, the interpolation with $\lambda = 0$ will be the best estimator since there is no noise.

## C  APPROXIMATION LEMMAS

In the following proofs, we always assume that $\delta \in (0, 1)$. For convenience, we use notations like $C, c$ to represent constants independent of $n, \delta$, which may vary from appearance to appearance.

### C.1  CONCENTRATION RESULTS

**Lemma C.1.** *Suppose that Assumption 1 holds. Let $f \in \mathcal{H}$ be given. We have*

$$\|(T_X - T)f\|_{\mathcal{H}} \leq 2\kappa \left(\frac{2\|f\|_\infty}{n} + \frac{\|f\|_{L^2}}{\sqrt{n}}\right) \ln \frac{2}{\delta} \tag{61}$$

*holds with probability at least $1 - \delta$.*

*Proof of Lemma C.1.* Let us define an $\mathcal{H}$-valued random variable $\xi_x = T_x f := K_x K_x^* f = f(x)k(x, \cdot)$. It is easy to verify that

$$\mathbb{E}_{x \sim \mu} \xi_x = Tf, \quad \text{and} \quad \frac{1}{n}\sum_{i=1}^{n} \xi_{x_i} = T_X f.$$

Furthermore, we have

$$\|\xi_x\|_{\mathcal{H}} = \|f(x)k(x, \cdot)\|_{\mathcal{H}} \leq \kappa\|f\|_\infty$$

and

$$\mathbb{E}_{x \sim \mu}\|\xi_x\|_{\mathcal{H}}^2 = \mathbb{E}_{x \sim \mu}\|f(x)k(x, \cdot)\|_{\mathcal{H}}^2 = \mathbb{E}_{x \sim \mu}|f(x)|^2 \kappa^2 = \kappa^2\|f\|_{L^2}^2.$$

Therefore, the proof is concluded by applying Lemma D.6 with $L = 2\kappa\|f\|_\infty$ and $\sigma = \kappa\|f\|_{L^2}$. $\square$

The following lemma shows that $(T_X + \lambda)^{-1}$ approximates to $(T + \lambda)^{-1}$. It is similar to Lin & Cevher (2020, Lemma 19), but here we do not require that $\lambda = n^{-\theta}$.

**Lemma C.2.** *Suppose that the Assumption 1 holds. If $n, \lambda$ satisfy*

$$\frac{\kappa^2}{\lambda n} \ln \frac{4\mathcal{N}(\lambda)}{\delta} \leq \frac{1}{16}, \tag{62}$$

*where $\mathcal{N}(\lambda) = \mathrm{Tr}(T + \lambda)^{-1}T$, then with probability at least $1 - \delta$ we have*

$$\left\|(T + \lambda)^{-\frac{1}{2}}(T_X + \lambda)^{\frac{1}{2}}\right\|_{\mathscr{B}(\mathcal{H})}^2, \left\|(T + \lambda)^{\frac{1}{2}}(T_X + \lambda)^{-\frac{1}{2}}\right\|_{\mathscr{B}(\mathcal{H})}^2 \leq 2. \tag{63}$$

*If Condition $(\mathbf{A})$ ( i.e, the eigen-value decay condition (9) ) holds and $\lambda = \Omega(n^{-(1-\varepsilon)})$ for some $\varepsilon \in (0, 1)$, then condition (62) holds for sufficiently large $n$.*

To prove Lemma C.2, we first prove the following lemma, which is a modified version of Lin & Cevher (2020, Lemma 16).

**Lemma C.3.** *Under Assumption* 1 *and condition* (9)*, the following holds with probability at least* $1 - \delta$*:*

$$\left\|(T+\lambda)^{-\frac{1}{2}}(T-T_X)(T+\lambda)^{-\frac{1}{2}}\right\| \leq \frac{4\kappa^2 B}{3\lambda n} + \sqrt{\frac{2\kappa^2 B}{\lambda n}}, \tag{64}$$

*where*

$$B = \ln \frac{4(\|T\| + \lambda)\mathcal{N}(\lambda)}{\delta \|T\|}.$$

*Proof.* We prove by using Lemma D.7. Let

$$A_i = A(x_i) = (T+\lambda)^{-\frac{1}{2}}(T_x - T)(T+\lambda)^{-\frac{1}{2}}.$$

Then, $\mathbb{E}A_i = 0$ and

$$\frac{1}{n}\sum_{i=1}^{n} A_i = (T+\lambda)^{-\frac{1}{2}}(T_x - T)(T+\lambda)^{-\frac{1}{2}}.$$

Calculation shows that

$$\|A\| \leq \left\|(T+\lambda)^{-\frac{1}{2}}\right\| (\|T_X\| + \|T\|) \left\|(T+\lambda)^{-\frac{1}{2}}\right\| \leq 2\kappa^2\lambda^{-1} = L.$$

Using the fact that $\mathbb{E}(B - \mathbb{E}B)^2 \preceq \mathbb{E}B^2$ for self-adjoint operator $B$, we have

$$\mathbb{E}A^2 \preceq \mathbb{E}\left[(T+\lambda)^{-\frac{1}{2}}T_x(T+\lambda)^{-\frac{1}{2}}\right]^2.$$

Moreover, noticing that $T \succeq 0$ and $0 \preceq T_x \preceq \kappa^2$, we have

$$A \preceq (T+\lambda)^{-\frac{1}{2}}\kappa^2(T+\lambda)^{-\frac{1}{2}} \preceq \kappa^2\lambda^{-1}$$

and hence

$$\mathbb{E}A^2 \preceq \kappa^2\lambda^{-1}\mathbb{E}\left[(T+\lambda)^{-\frac{1}{2}}T_x(T+\lambda)^{-\frac{1}{2}}\right] = \kappa^2\lambda^{-1}T(T+\lambda)^{-1} =: V.$$

We get

$$\|V\| = \kappa^2\lambda^{-1}\|T(T+\lambda)^{-1}\| = \kappa^2\lambda^{-1}\frac{\lambda_1}{\lambda + \lambda_1} = \kappa^2\lambda^{-1}\frac{\|T\|}{\|T\| + \lambda}$$

$$\mathrm{Tr}\, V = \kappa^2\lambda^{-1}\,\mathrm{Tr}\left[T(T+\lambda)^{-1}\right] = \kappa^2\lambda^{-1}\mathcal{N}(\lambda),$$

implying that

$$B = \ln\frac{4\,\mathrm{Tr}\, V}{\delta\|V\|} = \ln\frac{4(\|T\| + \lambda)\mathcal{N}(\lambda)}{\delta\|T\|},$$

and $\|V\| \leq \kappa^2\lambda^{-1}$. $\qquad\square$

Now we are ready to prove Lemma C.2.

*Proof of Lemma* C.2. Let

$$u = \frac{\kappa^2 B}{\lambda n} = \frac{\kappa^2}{\lambda n}\ln\frac{4(\|T\| + \lambda)\mathcal{N}(\lambda)}{\delta\|T\|}.$$

By Lemma C.3, with probability at least $1 - \delta$, we have

$$a = \left\|(T+\lambda)^{-\frac{1}{2}}(T-T_X)(T+\lambda)^{-\frac{1}{2}}\right\| \leq \frac{4}{3}u + \sqrt{2u} \leq \frac{1}{2},$$

where the last inequality comes from (62), namely $u \leq \frac{1}{16}$.

Then, for the first term we have

$$
\begin{aligned}
\left\| (T+\lambda)^{-\frac{1}{2}}(T_X+\lambda)^{\frac{1}{2}} \right\|^2 &= \left\| (T+\lambda)^{-1/2}(T_X+\lambda)(T+\lambda)^{-1/2} \right\| \\
&= \left\| (T+\lambda)^{-1}(T_X - T + T + \lambda)(T+\lambda)^{-1} \right\| \\
&= \left\| (T+\lambda)^{-1}(T_X - T)(T+\lambda)^{-1} + I \right\| \\
&\leq a + 1 \leq 2.
\end{aligned}
$$

Similarly, the second term can be bounded by

$$
\begin{aligned}
\left\| (T+\lambda)^{\frac{1}{2}}(T_X+\lambda)^{-\frac{1}{2}} \right\|^2 &= \left\| (T+\lambda)^{1/2}(T_X+\lambda)^{-1}(T+\lambda)^{1/2} \right\| \\
&= \left\| \left[ (T+\lambda)^{-1/2}(T_X+\lambda)(T+\lambda)^{-1/2} \right]^{-1} \right\| \\
&\leq (1-a)^{-1} \leq 2.
\end{aligned}
$$

Furthermore, if condition (9) holds and $\lambda = \Omega(n^{-(1-\varepsilon)})$, then from Proposition D.1 we get

$$
\mathcal{N}(\lambda) \leq C\lambda^{-\beta} = O\left( n^{\beta(1-\varepsilon)} \right).
$$

Therefore,

$$
\frac{\kappa^2}{\lambda n} \ln \frac{4(\|T\| + \lambda)\mathcal{N}(\lambda)}{\delta \|T\|} \leq C n^{-\varepsilon} \left( (1+\beta)(1-\varepsilon)\ln n + \ln \frac{4}{\delta} + C \right) \to 0 \quad \text{as} \quad n \to \infty.
$$

$\square$

## C.2 Norm control of regularized functions

**Proposition C.4.** *Suppose that $f \in [\mathcal{H}]^\alpha$. Then, for any $0 \leq \gamma \leq \alpha$ such that $\alpha - \gamma \leq 2$, we have*

$$
\left\| (T+\lambda)^{-1}f \right\|_{[\mathcal{H}]^\gamma} \leq \lambda^{\frac{\alpha-\gamma}{2}-1}\|f\|_{[\mathcal{H}]^\alpha}. \tag{65}
$$

*Proof.* From the definition of $\|\cdot\|_{[\mathcal{H}]^\gamma}$, we have $f = T^{\alpha/2}f_0$ for some $f_0 \in L^2$ such that $\|f_0\|_{L^2} = \|f\|_{[\mathcal{H}]^\alpha}$, and

$$
\begin{aligned}
\left\| (T+\lambda)^{-1}f \right\|_{[\mathcal{H}]^\gamma} &= \left\| T^{-\gamma/2}(T+\lambda)^{-1}T^{\alpha/2}f_0 \right\|_{L^2} \\
&= \left\| T^{\frac{\alpha-\gamma}{2}}(T+\lambda)^{-1}f_0 \right\|_{L^2} \\
&\leq \left\| T^{\frac{\alpha-\gamma}{2}}(T+\lambda)^{-1} \right\|_{\mathscr{B}(L^2)} \|f_0\|_{L^2} \\
&\leq \lambda^{\frac{\alpha-\gamma}{2}-1}\|f\|_{[\mathcal{H}]^\alpha},
\end{aligned}
$$

where the last inequality comes from applying Proposition D.2 to operator calculus. $\square$

The following special cases of Proposition C.4 are useful in our proofs. We present them as corollaries. Notice that we have (29), so from estimations of the RKHS-norm we can also get estimations of the sup-norm.

**Corollary C.5.** *For $f \in [\mathcal{H}]^2$, we have the following estimations:*

$$
\left\| (T+\lambda)^{-1}f \right\|_{L^2} \leq \|f\|_{[\mathcal{H}]^2},
$$

$$
\left\| (T+\lambda)^{-1}f \right\|_\infty \leq \kappa\lambda^{-1/2}\|f\|_{[\mathcal{H}]^2}.
$$

*Proof.* Applying Proposition C.4 with $\alpha = 2$ and $\gamma = 0, 1$ respectively, we obtain

$$
\left\| (T+\lambda)^{-1}f \right\|_{L^2} \leq \|f\|_{[\mathcal{H}]^2},
$$

$$
\left\| (T+\lambda)^{-1}f \right\|_\infty \leq \kappa \left\| (T+\lambda)^{-1}f \right\|_\mathcal{H} \leq \kappa\lambda^{-1/2}\|f\|_{[\mathcal{H}]^2}.
$$

$\square$

Similarly, noticing that $k(x, \cdot) \in [\mathcal{H}]^1$, we also have the following corollary controlling the norms of regularized kernel basis function $(T + \lambda)^{-1} k(x, \cdot)$:

**Corollary C.6.** *We have the following estimations:* $\forall x \in \mathcal{X}$,

$$
\left\| (T + \lambda)^{-1} k(x, \cdot) \right\|_{L^2} \leq \kappa \lambda^{-1/2}, \quad \left\| (T + \lambda)^{-1} k(x, \cdot) \right\|_{\mathcal{H}} \leq \kappa \lambda^{-1},
$$
$$
\left\| (T + \lambda)^{-1} k(x, \cdot) \right\|_{\infty} \leq \kappa^2 \lambda^{-1}.
$$

(66)

*where $C$ is a positive constant.*

### C.3 Approximation of the Regularized Regression Function

**Lemma C.7.** *Suppose that $f^* \in [\mathcal{H}]^2$. If $\lambda = \lambda(n) = \Omega(n^{-(1-\varepsilon)})$ for some $\varepsilon > 0$ and $\lambda(n) \to 0$, then there exist some $q > 0$ such that for sufficient large $n$, the following holds with probability at least $1 - \delta$:*

$$
\left| \left\| (T + \lambda)^{-1} f^* \right\|_{L^2} - \left\| (T_X + \lambda)^{-1} f^* \right\|_{L^2} \right| = O(n^{-q}) \ln \frac{4}{\delta}.
$$

(67)

*Proof.* By the triangle inequality and noticing that

$$
(T + \lambda)^{-1} - (T_X + \lambda)^{-1} = (T_X + \lambda)^{-1}(T_X - T)(T + \lambda)^{-1},
$$

we have

$$
\left| \left\| (T + \lambda)^{-1} f^* \right\|_{L^2} - \left\| (T_X + \lambda)^{-1} f^* \right\|_{L^2} \right|
$$
$$
\leq \left\| \left( (T + \lambda)^{-1} - (T_X + \lambda)^{-1} \right) f^* \right\|_{L^2}
$$
$$
= \left\| T^{1/2}(T_X + \lambda)^{-1}(T_X - T)(T + \lambda)^{-1} f^* \right\|_{\mathcal{H}}
$$
$$
\leq \left\| T^{1/2}(T_X + \lambda)^{-1} \right\|_{\mathscr{B}(\mathcal{H})} \left\| (T_X - T)(T + \lambda)^{-1} f^* \right\|_{\mathcal{H}}.
$$

(68)

For the first term in (68), we have

$$
\left\| T^{1/2}(T_X + \lambda)^{-1} \right\|_{\mathscr{B}(\mathcal{H})} = \left\| T^{1/2}(T + \lambda)^{-1}(T + \lambda)(T_X + \lambda)^{-1} \right\|_{\mathscr{B}(\mathcal{H})}
$$
$$
\leq \left\| T^{1/2}(T + \lambda)^{-1} \right\|_{\mathscr{B}(\mathcal{H})} \left\| (T + \lambda)(T_X + \lambda)^{-1} \right\|_{\mathscr{B}(\mathcal{H})}
$$
$$
\text{(By Proposition D.2)} \quad \leq \lambda^{-1/2} \left\| (T + \lambda)(T_X + \lambda)^{-1} \right\|_{\mathscr{B}(\mathcal{H})}.
$$

Since $\lambda = \Omega(n^{-(1-\varepsilon)})$, Lemma C.2 yields that

$$
\left\| (T + \lambda)^{1/2}(T_X + \lambda)^{-1/2} \right\|_{\mathscr{B}(\mathcal{H})} \leq 2
$$

holds with probability at least $1 - \delta/2$ for sufficient large $n$. Because the operator $(T + \lambda)^{1/2}(T_X + \lambda)^{-1/2}$ is invertible, we get

$$
\left\| (T_X + \lambda)^{1/2}(T + \lambda)^{-1/2} \right\|_{\mathscr{B}(\mathcal{H})} \geq \frac{1}{2}.
$$

By Lemma D.4, we have

$$
\left\| (T_X + \lambda)(T + \lambda)^{-1} \right\|_{\mathscr{B}(\mathcal{H})} \geq \left\| (T_X + \lambda)^{1/2}(T + \lambda)^{-1/2} \right\|_{\mathscr{B}(\mathcal{H})}^2 \geq \frac{1}{4},
$$

which implies that

$$
\left\| (T + \lambda)(T_X + \lambda)^{-1} \right\|_{\mathscr{B}(\mathcal{H})} \leq 4.
$$

Therefore, we obtain the upper bound

$$
\left\| T^{1/2}(T_X + \lambda)^{-1} \right\|_{\mathscr{B}(\mathcal{H})} \leq 4\lambda^{-1/2}.
$$

(69)

For the second term in (68), we apply Lemma C.1 with $f = (T + \lambda)^{-1}f^*$ and get that

$$\left\|(T_X - T)(T + \lambda)^{-1}f^*\right\|_{\mathcal{H}} \leq 2\kappa \left( \frac{2\left\|(T + \lambda)^{-1}f^*\right\|_\infty}{n} + \frac{\left\|(T + \lambda)^{-1}f^*\right\|_{L^2}}{\sqrt{n}} \right) \ln \frac{4}{\delta}$$

holds with probability at least $1 - \delta/2$. Plugging in the bounds from Corollary C.5, we obtain that

$$\left\|(T_X - T)(T + \lambda)^{-1}f^*\right\|_{\mathcal{H}} \leq C \left( \frac{\lambda^{-1/2}}{n} + \frac{1}{\sqrt{n}} \right) \ln \frac{4}{\delta}. \tag{70}$$

Plugging (69) and (70) back into (68), we finally get

$$\left| \left\|(T + \lambda)^{-1}f^*\right\|_{L^2} - \left\|(T_X + \lambda)^{-1}f^*\right\|_{L^2} \right| \leq C \left( \frac{\lambda^{-1}}{n} + \frac{\lambda^{-1/2}}{\sqrt{n}} \right) \ln \frac{4}{\delta} = O(n^{-q}) \ln \frac{4}{\delta},$$

where we use the condition $\lambda = \Omega(n^{-(1-\varepsilon)})$ in the last equality. $\qquad\square$

Combining the previous lemma with $L^2$-norm control of the regularized regression function, we obtain the following corollary:

**Corollary C.8.** *Suppose that $f^* \in [\mathcal{H}]^2$. If $\lambda = \lambda(n) = \Omega(n^{-(1-\varepsilon)})$ for some $\varepsilon > 0$ and $\lambda(n) \to 0$, for sufficient large $n$, the following holds with probability at least $1 - \delta$:*

$$\left| \left\|(T + \lambda)^{-1}f^*\right\|_{L^2}^2 - \left\|(T_X + \lambda)^{-1}f^*\right\|_{L^2}^2 \right| = O(n^{-q}) \left( \ln \frac{4}{\delta} \right)^2. \tag{71}$$

*Proof.* From Lemma C.7 and $\left\|(T + \lambda)^{-1}f^*\right\|_{L^2} \leq \|f^*\|_{[\mathcal{H}]^2}$ in Corollary C.5, we have

$$\left\|(T_X + \lambda)^{-1}f^*\right\|_{L^2} \leq \left\|(T + \lambda)^{-1}f^*\right\|_{L^2} + \left| \left\|(T + \lambda)^{-1}f^*\right\|_{L^2} - \left\|(T_X + \lambda)^{-1}f^*\right\|_{L^2} \right|$$

$$\leq \|f^*\|_{[\mathcal{H}]^2} + o(1) \ln \frac{4}{\delta} = O(1) \ln \frac{4}{\delta},$$

and hence

$$\left\|(T + \lambda)^{-1}f^*\right\|_{L^2} + \left\|(T_X + \lambda)^{-1}f^*\right\|_{L^2} = O(1) \ln \frac{4}{\delta}.$$

Therefore, we get

$$\left| \left\|(T + \lambda)^{-1}f^*\right\|_{L^2}^2 - \left\|(T_X + \lambda)^{-1}f^*\right\|_{L^2}^2 \right|$$

$$= \left| \left\|(T + \lambda)^{-1}f^*\right\|_{L^2} - \left\|(T_X + \lambda)^{-1}f^*\right\|_{L^2} \right| \cdot \left( \left\|(T + \lambda)^{-1}f^*\right\|_{L^2} + \left\|(T_X + \lambda)^{-1}f^*\right\|_{L^2} \right)$$

$$= O(n^{-q}) \ln \frac{4}{\delta} \cdot O(1) \ln \frac{4}{\delta}$$

$$= O(n^{-q}) \left( \ln \frac{4}{\delta} \right)^2.$$

$\qquad\square$

## C.4 Approximation of the regularized kernel basis function

The following proposition about estimating the $L^2$ norm with empirical norms is a corollary of Lemma D.5.

**Proposition C.9.** *Let $\mu$ be a probability measure on $\mathcal{X}$, $f \in L^2(\mathcal{X}, \mathrm{d}\mu)$ and $\|f\|_\infty \leq M$. Suppose we have $x_1, \ldots, x_n$ sampled i.i.d. from $\mu$. Then, for any $\alpha > 0$, the following holds with probability at least $1 - \delta$:*

$$\left| \|f\|_{L^2,n}^2 - \|f\|_{L^2}^2 \right| \leq \alpha M^2 \|f\|_{L^2}^2 + \frac{3 + 4\alpha M^2}{6\alpha n} \ln \frac{2}{\delta}.$$

*By choosing $\alpha = \frac{1}{2M^2}$, we have*

$$\frac{1}{2} \|f\|_{L^2}^2 - \frac{5M^2}{3n} \ln \frac{2}{\delta} \leq \|f\|_{L^2,n}^2 \leq \frac{3}{2} \|f\|_{L^2}^2 + \frac{5M^2}{3n} \ln \frac{2}{\delta}. \tag{72}$$

*Proof.* Defining $\xi_i = f(x_i)^2$, we have

$$\mathbb{E}\xi_i = \|f\|_{L^2}^2,$$
$$\mathbb{E}\xi_i^2 = \mathbb{E}_{x \sim \mu} f(x)^4 \leq \|f\|_\infty^2 \|f\|_{L^2}^2.$$

Therefore, applying Lemma D.5, we get

$$\left| \|f\|_{L^2,n}^2 - \|f\|_{L^2}^2 \right| \leq \alpha \|f\|_\infty^2 \|f\|_{L^2}^2 + \frac{3 + 4\alpha M^2}{6\alpha n} \ln \frac{2}{\delta}.$$

$\square$

We establish the following lemma about covering numbers of the regularized kernel basis functions. For simplicity, let us denote $h_x = k(x, \cdot) \in \mathcal{H}$ and

$$\mathcal{K}_\lambda := \left\{ (T + \lambda)^{-1} h_x \right\}_{x \in \mathcal{X}}. \tag{73}$$

**Lemma C.10.** *Assuming that $\mathcal{X} \subseteq \mathbb{R}^d$ is bounded and $k \in C^s(\mathcal{X} \times \mathcal{X})$ for some $s \in (0, 1]$. Then, we have*

$$\mathcal{N}\left(\mathcal{K}_\lambda, \|\cdot\|_\infty, \varepsilon\right) \leq C \left(\lambda\varepsilon\right)^{-\frac{2d}{s}}, \tag{74}$$

$$\mathcal{N}\left(\mathcal{K}_\lambda, \|\cdot\|_\mathcal{H}, \varepsilon\right) \leq C \left(\lambda^{1+\frac{\beta}{2}}\varepsilon\right)^{-\frac{2d}{s}}, \tag{75}$$

*where $C$ is a positive constant not depending on $\lambda$ or $\varepsilon$.*

*Proof.* We first prove (74). By Mercer's theorem, we have

$$(T + \lambda)^{-1} h_a = \sum_{i \in N} \frac{\lambda_i}{\lambda + \lambda_i} e_i(a) e_i,$$

and thus

$$\left[(T + \lambda)^{-1} h_a\right](x) = \sum_{i \in N} \frac{\lambda_i}{\lambda + \lambda_i} e_i(a) e_i(x)$$
$$= \left[(T + \lambda)^{-1} h_x\right](a).$$

Therefore,

$$\left\|(T + \lambda)^{-1} h_a - (T + \lambda)^{-1} h_b\right\|_\infty = \sup_{x \in \mathcal{X}} \left| \left[(T + \lambda)^{-1} h_a\right](x) - \left[(T + \lambda)^{-1} h_b\right](x) \right|$$
$$= \sup_{x \in \mathcal{X}} \left| \left[(T + \lambda)^{-1} h_x\right](a) - \left[(T + \lambda)^{-1} h_x\right](b) \right|.$$

Since $k$ is Hölder-continuous, by Lemma A.3 we know that $(T + \lambda)^{-1} h_x$ is also Hölder-continuous. Plugging the bound $\left\|(T + \lambda)^{-1} h_x\right\| \leq \kappa\lambda^{-1}$ obtained in Corollary C.6 into (31), we get

$$[(T + \lambda)^{-1} h_x]_{s/2,\mathcal{X}} \leq \sqrt{2\kappa^2 [k]_{s,\mathcal{X} \times \mathcal{X}}} \left\|(T + \lambda)^{-1} h_x\right\|_\mathcal{H} \leq \kappa^2 \sqrt{2[k]_{s,\mathcal{X} \times \mathcal{X}}} \lambda^{-1},$$

which implies that

$$\left| \left[(T + \lambda)^{-1} h_x\right](a) - \left[(T + \lambda)^{-1} h_x\right](b) \right| \leq C_0 \lambda^{-1} |a - b|^{s/2},$$

where $C_0 = \kappa^2 \sqrt{2[k]_{s,\mathcal{X} \times \mathcal{X}}}$. Consequently, we have

$$\left\|(T + \lambda)^{-1} h_a - (T + \lambda)^{-1} h_b\right\|_\infty \leq C_0 \lambda^{-1} |a - b|^{s/2}. \tag{76}$$

(76) yields that to find an $\varepsilon$-net of $\mathcal{K}_\lambda$ with respect to $\|\cdot\|_\infty$, we only need to find an $\tilde{\varepsilon}$-net of $\mathcal{X}$ with respect to the Euclidean norm, where $\tilde{\varepsilon} = \left(\frac{\lambda\varepsilon}{C_0}\right)^{2/s}$. Since the result of the covering number of the latter one is already known in Lemma A.8, we finally obtain that

$$\mathcal{N}\left(\mathcal{K}_\lambda, \|\cdot\|_\infty, \varepsilon\right) \leq \mathcal{N}\left(\mathcal{X}, \|\cdot\|_{\mathbb{R}^d}, \tilde{\varepsilon}\right) \leq C\tilde{\varepsilon}^{-d} = C \left(\frac{\lambda\varepsilon}{C_0}\right)^{-\frac{2d}{s}} = \tilde{C} \left(\lambda\varepsilon\right)^{-\frac{2d}{s}}.$$

Now we prove (75). By Mercer's theorem and the series definition of the RHKS norm, we have

$$\left\|(T+\lambda)^{-1}h_x - (T+\lambda)^{-1}h_y\right\|_{\mathcal{H}}^2 = \left\|\sum_{i\in N}\frac{\lambda_i\left(e_i(x)-e_i(y)\right)}{\lambda+\lambda_i}e_i\right\|_{\mathcal{H}}^2$$

$$= \sum_{i\in N}\frac{\lambda_i\left(e_i(x)-e_i(y)\right)^2}{(\lambda+\lambda_i)^2}.$$

Since $\|e_i\|_{\mathcal{H}} = \lambda_i^{-1/2}$, by Lemma A.3 we have

$$[e_i]_{s/2,\mathcal{X}} \le \kappa^2\sqrt{2[k]_{s,\mathcal{X}\times\mathcal{X}}}\lambda_i^{-1/2},$$

and thus $(e_i(x)-e_i(y))^2 \le C\lambda_i^{-1}|x-y|^s$. Therefore, we get

$$\left\|(T+\lambda)^{-1}h_x - (T+\lambda)^{-1}h_y\right\|_{\mathcal{H}}^2 \le C\sum_{i\in N}\frac{|x-y|^s}{(\lambda+\lambda_i)^2}$$

$$= C|x-y|^s\lambda^{-2}\sum_{i\in N}\left(\frac{\lambda}{\lambda+\lambda_i}\right)^2$$

$$\le C|x-y|^s\lambda^{-(2+\beta)}.$$

Using a similar covering argument over $\mathcal{X}$ gives the desired result. $\qquad\square$

**Lemma C.11.** *Suppose that Assumption 1 holds. Assume that $\lambda = \lambda(n) = \Omega(n^{-1/2+p})$ for some $p \in (0,1/2)$. Then, there exists some $q > 0$ such that for any $\delta > 0$, the following holds with probability at least $1-\delta$: $\forall x \in \mathcal{X}$,*

$$\left\|(T+\lambda)^{-1}h_x\right\|_{L^2,n}^2 \le \frac{3}{2}\left\|(T+\lambda)^{-1}h_x\right\|_{L^2}^2 + O(n^{-q})\ln\frac{2}{\delta},$$

$$\left\|(T+\lambda)^{-1}h_x\right\|_{L^2,n}^2 \ge \frac{1}{2}\left\|(T+\lambda)^{-1}h_x\right\|_{L^2}^2 - O(n^{-q})\ln\frac{2}{\delta},$$

*where the constants in $O(n^{-q})$ do not depend on $x$.*

*Proof.* By Lemma C.10, we can find an $\varepsilon$-net $\mathcal{F} \subseteq \mathcal{K}_\lambda \subseteq \mathcal{H}$ with respect to sup-norm of $\mathcal{K}_\lambda$ such that

$$|\mathcal{F}| \le C\left(\lambda\varepsilon\right)^{-\frac{2d}{s}}, \tag{77}$$

where $\varepsilon = \varepsilon(n)$ will be determined later.

Applying Proposition C.9 to $\mathcal{F}$ with the $\|\cdot\|_\infty$-bound in Corollary C.6, with probability at least $1-\delta$ we have

$$\frac{1}{2}\|f\|_{L^2}^2 - \frac{C\lambda^{-2}}{n}\ln\frac{2|\mathcal{F}|}{\delta} \le \|f\|_{L^2,n}^2 \le \frac{3}{2}\|f\|_{L^2}^2 + \frac{C\lambda^{-2}}{n}\ln\frac{2|\mathcal{F}|}{\delta}, \quad \forall f \in \mathcal{F}. \tag{78}$$

Now, since $\mathcal{F}$ is an $\varepsilon$-net of $\mathcal{K}_\lambda$ with respect to $\|\cdot\|_\infty$, for any $x \in \mathcal{X}$, there exists some $f \in \mathcal{F}$ such that

$$\left\|(T+\lambda)^{-1}h_x - f\right\|_\infty \le \varepsilon,$$

which implies that

$$\left|\left\|(T+\lambda)^{-1}h_x\right\|_{L^2} - \|f\|_{L^2}\right| \le \varepsilon, \quad \left|\left\|(T+\lambda)^{-1}h_x\right\|_{L^2,n} - \|f\|_{L^2,n}\right| \le \varepsilon.$$

Since $\left\|(T+\lambda)^{-1}h_x\right\|_\infty \le C\lambda^{-1}$ and $a^2 - b^2 = (a-b)(2b+(a-b))$, we get

$$\left|\left\|(T+\lambda)^{-1}h_x\right\|_{L^2}^2 - \|f\|_{L^2}^2\right| \le C\varepsilon\lambda^{-1}, \quad \left|\left\|(T+\lambda)^{-1}h_x\right\|_{L^2,n}^2 - \|f\|_{L^2,n}^2\right| \le C\varepsilon\lambda^{-1}. \tag{79}$$

For the upper bound, we have

$$\left\|(T+\lambda)^{-1}h_x\right\|_{L^2,n}^2 \le \|f\|_{L^2,n}^2 + C\varepsilon\lambda^{-1} \quad \text{(By (79))}$$

$$\text{(By (78))} \quad \leq \frac{3}{2}\|f\|_{L^2}^2 + \frac{C\lambda^{-2}}{n}\ln\frac{2|\mathcal{F}|}{\delta} + C\varepsilon\lambda^{-1}$$

$$\text{(By (79) again)} \quad \leq \frac{3}{2}\left\|(T+\lambda)^{-1}h_x\right\|_{L^2}^2 + \frac{C\lambda^{-2}}{n}\ln\frac{2|\mathcal{F}|}{\delta} + C\varepsilon\lambda^{-1}.$$

Noticing that $\lambda = \Omega\left(n^{-1/2+p}\right)$ and (77), by choosing $\varepsilon = n^{-1}$, it is easy to verify that

$$\frac{C\lambda^{-2}}{n}\ln\frac{2|\mathcal{F}|}{\delta} + C\varepsilon\lambda^{-1} = O\left(n^{-2p}\right)\left(\ln|\mathcal{F}| + \ln\frac{2}{\delta}\right) + O(n^{-\frac{1}{2}-p})$$

$$= O\left(n^{-2p}\right)\left(C_1\ln n + C_2 + \ln\frac{2}{\delta}\right) + O(n^{-\frac{1}{2}-p})$$

$$= O\left(n^{-q}\right)\ln\frac{2}{\delta}$$

for some $q > 0$. The lower bound follows similarly. $\qquad\square$

**Lemma C.12.** *Suppose that Assumption 1 and Condition* $(\mathbf{A})$ *( i.e., the eigenvalue decay rate (9) ) hold. Assume that* $\lambda = \Omega\left(n^{-\frac{1}{2}+p}\right)$ *for some* $p \in (0, 1/2)$. *Then, there exists some* $q > 0$ *such that*

$$\left|\left\|T_X^{1/2}(T_X + \lambda)^{-1}h_x\right\|_{\mathcal{H}} - \left\|T_X^{1/2}(T+\lambda)^{-1}h_x\right\|_{\mathcal{H}}\right| \leq O(n^{-q})\ln\frac{2}{\delta}, \quad \forall x \in \mathcal{X},$$

*with probability at least* $1 - \delta$, *where the constant in* $O(n^{-q})$ *is independent of* $x$.

*Proof.* We begin with

$$\left|\left\|T_X^{1/2}(T_X + \lambda)^{-1}h_x\right\|_{\mathcal{H}} - \left\|T_X^{1/2}(T+\lambda)^{-1}h_x\right\|_{\mathcal{H}}\right| \leq \left\|T_X^{1/2}\left[(T_X + \lambda)^{-1} - (T+\lambda)^{-1}\right]h_x\right\|_{\mathcal{H}}.$$

Noticing that

$$(T_X + \lambda)^{-1} - (T+\lambda)^{-1} = (T_X + \lambda)^{-1}(T - T_X)(T+\lambda)^{-1},$$

we obtain

$$\left\|T_X^{1/2}\left[(T_X + \lambda)^{-1} - (T+\lambda)^{-1}\right]h_x\right\| = \left\|T_X^{1/2}(T_X + \lambda)^{-1}(T - T_X)(T+\lambda)^{-1}h_x\right\|_{\mathcal{H}}$$

$$\leq \left\|T_X^{1/2}(T_X + \lambda)^{-1}\right\|\left\|(T - T_X)(T+\lambda)^{-1}h_x\right\|_{\mathcal{H}}$$

$$\leq \lambda^{-1/2}\left\|(T - T_X)(T+\lambda)^{-1}h_x\right\|_{\mathcal{H}} \tag{80}$$

where we apply Proposition D.2 in the last inequality.

Now, we deal with the last term in (80). By Lemma C.10, we find an $\varepsilon$-covering $\mathcal{F}$ of $\mathcal{K}_\lambda$ with respect to $\|\cdot\|_{\mathcal{H}}$ such that

$$|\mathcal{F}| \leq C\left(\lambda^{\frac{2+\beta}{2}}\varepsilon\right)^{-\frac{2d}{s}}. \tag{81}$$

Applying Lemma C.1 to $\mathcal{F}$ and noticing the $L^2$-norm and sup-norm bounds obtained in Corollary C.6, we find that $\forall f \in \mathcal{F}$,

$$\|(T - T_X)f\|_{\mathcal{H}} \leq 2\kappa\left(\frac{2\|f\|_\infty}{n} + \frac{\|f\|_{L^2}}{\sqrt{n}}\right)\ln\frac{2|\mathcal{F}|}{\delta}$$

$$\leq C\left(\frac{\lambda^{-1}}{n} + \frac{\lambda^{-1/2}}{\sqrt{n}}\right)\ln\frac{2|\mathcal{F}|}{\delta}.$$

Then, for any $x \in \mathcal{X}$, we can find some $f \in \mathcal{F}$ such that $\left\|(T+\lambda)^{-1}h_x - f\right\|_{\mathcal{H}} \leq \varepsilon$, and thus

$$\left\|(T - T_X)(T+\lambda)^{-1}h_x\right\|_{\mathcal{H}} \leq \left\|(T - T_X)\left((T+\lambda)^{-1}h_x - f\right)\right\|_{\mathcal{H}} + \|(T - T_X)f\|_{\mathcal{H}}$$

$$\leq \|T - T_X\|_{\mathcal{B}(\mathcal{H})}\left\|(T+\lambda)^{-1}h_x - f\right\|_{\mathcal{H}} + \|(T - T_X)f\|_{\mathcal{H}}$$

$$\leq 2\kappa^2\varepsilon + C\left(\frac{\lambda^{-1}}{n} + \frac{\lambda^{-1/2}}{\sqrt{n}}\right)\ln\frac{2|\mathcal{F}|}{\delta}. \tag{82}$$

Plugging (82) into (80), we obtain

$$\left\| T_X^{1/2} \left[ (T_X + \lambda)^{-1} - (T + \lambda)^{-1} \right] h_x \right\| \leq C\lambda^{-1/2}\varepsilon + C\left( \frac{\lambda^{-\frac{3}{2}}}{n} + \frac{\lambda^{-1}}{\sqrt{n}} \right) \ln \frac{2|\mathcal{F}|}{\delta}.$$

Finally, noticing $\lambda = \Omega\left( n^{-\frac{1}{2}+p} \right)$ and (81), by letting $\varepsilon = n^{-1}$ we have

$$\left\| T_X^{1/2} \left[ (T_X + \lambda)^{-1} - (T + \lambda)^{-1} \right] h_x \right\| \leq Cn^{-q} \ln \frac{2}{\delta}$$

for some $q > 0$. $\qquad\square$

## D  AUXILIARY RESULTS

For any $p \geq 1$, let us introduce the $p$-effective dimension

$$\mathcal{N}_p(\lambda) := \mathrm{Tr}\left( T(T+\lambda)^{-1} \right)^p = \sum_{i=1}^{\infty} \left( \frac{\lambda_i}{\lambda + \lambda_i} \right)^p, \quad p \geq 1.$$

**Proposition D.1.** *If $\lambda_i \asymp i^{-1/\beta}$, we have*

$$\mathcal{N}_p(\lambda) \asymp \lambda^{-\beta}. \tag{83}$$

*Proof.* Since $c\, i^{-1/\beta} \leq \lambda_i \leq Ci^{-1/\beta}$, we have

$$\mathcal{N}_p(\lambda) = \sum_{i=1}^{\infty} \left( \frac{\lambda_i}{\lambda_i + \lambda} \right)^p \leq \sum_{i=1}^{\infty} \left( \frac{Ci^{-1/\beta}}{Ci^{-1/\beta} + \lambda} \right)^p = \sum_{i=1}^{\infty} \left( \frac{C}{C + \lambda i^{1/\beta}} \right)^p$$

$$\leq \int_0^{\infty} \left( \frac{C}{\lambda x^{1/\beta} + C} \right)^p \mathrm{d}x = \lambda^{-\beta} \int_0^{\infty} \left( \frac{C}{y^{1/\beta} + C} \right)^p \mathrm{d}y \leq \tilde{C}\lambda^{-\beta}.$$

for some constant $C$. Similarly, we hace

$$\mathcal{N}_p(\lambda) \geq \tilde{C}'\lambda^{-\beta}.$$

for some constant $\tilde{C}'$. $\qquad\square$

**Proposition D.2.** *For $\lambda > 0$ and $s \in [0, 1]$, we have*

$$\sup_{t \geq 0} \frac{t^s}{t + \lambda} \leq \lambda^{s-1}.$$

*Proof.* It follows from the inequality $a^s \leq a + 1$ for any $a \geq 0$ and $s \in [0, 1]$. $\qquad\square$

**Lemma D.3.** *(Young's inequality) Let $a, b > 0$. For $p, q > 1$ satisfying $\frac{1}{p} + \frac{1}{q} = 1$, we have*

$$ab \leq \frac{1}{p}a^p + \frac{1}{q}b^q, \tag{84}$$

*or equivalently*

$$a + b \geq \sqrt[p]{p}\sqrt[q]{q} \cdot a^{\frac{1}{p}} b^{\frac{1}{q}}. \tag{85}$$

The following operator inequality(Fujii et al., 1993) will be used in our proofs.

**Lemma D.4** (Cordes' Inequality). *Let $A, B$ be two positive semi-definite bounded linear operators on separable Hilbert space $H$. Then*

$$\|A^s B^s\|_{\mathscr{B}(H)} \leq \|AB\|_{\mathscr{B}(H)}^s, \quad \forall s \in [0, 1]. \tag{86}$$

### D.1 CONCENTRATION INEQUALITIES

The following concentration inequality is adopted from Caponnetto & Yao (2010):

**Lemma D.5.** *Let $\xi_1, \ldots, \xi_n$ be $n$ i.i.d. bounded random variables such that $|\xi_i| \leq B$, $\mathbb{E}\xi_i = \mu$, and $\mathbb{E}(\xi_i - \mu)^2 \leq \sigma^2$. Then for any $\alpha > 0$, any $\delta \in (0, 1)$, we have*

$$\left| \frac{1}{n} \sum_{i=1}^{n} \xi_i - \mu \right| \leq \alpha\sigma^2 + \frac{3 + 4\alpha B}{6\alpha n} \ln \frac{2}{\delta} \tag{87}$$

*holds with probability at least $1 - \delta$.*

*Proof of Lemma D.5.* The high probability form of bound in (87) is equivalent to the following probability form:

$$\mathbb{P}\left\{ \left| \frac{1}{n} \sum_{i=1}^{n} \xi_i - \mu \right| \geq \alpha\sigma^2 + \varepsilon \right\} \leq 2\exp\left( -\frac{6n\alpha\varepsilon}{3 + 4\alpha B} \right),$$

where $\varepsilon = \frac{3 + 4\alpha B}{6\alpha n} \ln \frac{2}{\delta}$. By symmetry, it suffices to prove the following one-sided inequality:

$$\mathbb{P}\left\{ \frac{1}{n} \sum_{i=1}^{n} \xi_i - \mu \geq \alpha\sigma^2 + \varepsilon \right\} \leq \exp\left( -\frac{6n\alpha\varepsilon}{3 + 4\alpha B} \right).$$

Taking exponent with some factor $s > 0$, we obtain

$$\mathbb{P}\left\{ \frac{1}{n} \sum_{i=1}^{n} \xi_i - \mu \geq \alpha\sigma^2 + \varepsilon \right\} = \mathbb{P}\left\{ \exp\left( \frac{s}{n} \sum_{i=1}^{n} (\xi_i - \mu) \right) \geq \exp\left( s(\alpha\sigma^2 + \varepsilon) \right) \right\}$$

$$(\text{Markov Inequality}) \quad \leq \exp\left( -s(\alpha\sigma^2 + \varepsilon) \right) \mathbb{E} \exp\left( \frac{s}{n} \sum_{i=1}^{n} (\xi_i - \mu) \right)$$

$$(\text{Independency}) \quad \leq \exp\left( -s(\alpha\sigma^2 + \varepsilon) \right) \prod_{i=1}^{n} \mathbb{E} \exp\left( \frac{s}{n} (\xi_i - \mu) \right) \tag{88}$$

Let $X_i = \xi_i - \mu$, $t = \frac{s}{n}$ and $M = 2B$. As long as $t < \frac{3}{M}$, we have

$$\mathbb{E} \exp\left( \frac{s}{n} (\xi_i - \mu) \right) = \mathbb{E} \exp(tX_i) = \sum_{k=0}^{\infty} \frac{t^k}{k!} \mathbb{E} X_i^k$$

$$\leq 1 + \sum_{k=2}^{\infty} \frac{t^k}{k!} M^{k-2}\sigma^2$$

$$\leq 1 + \frac{t^2\sigma^2}{2} \sum_{k=0}^{\infty} \left( \frac{Mt}{3} \right)^k$$

$$= 1 + \frac{3t^2\sigma^2}{6 - 2Mt}$$

$$\leq \exp\left( \frac{3t^2\sigma^2}{6 - 2Mt} \right).$$

Therefore,

$$(88) \leq \exp\left( -s(\alpha\sigma^2 + \varepsilon) + n\frac{3t^2\sigma^2}{6 - 2Mt} \right) = \exp\left( -s(\alpha\sigma^2 + \varepsilon) + \frac{3s^2\sigma^2}{6n - 4Bs} \right)$$

$$= \exp\left( -s\varepsilon + s\sigma^2 \left( \alpha + \frac{3s}{6n - 4Bs} \right) \right)$$

Solving $\alpha + \frac{3s}{6n - 4Bs} = 0$ and we get $s = \frac{6\alpha n}{3 + 4\alpha B}$, which satisfies that $t = \frac{s}{n} < \frac{3}{M}$, hence we have

$$(88) \leq \exp\left( -\frac{6\alpha n}{3 + 4\alpha B} \varepsilon \right)$$

and the proof is complete. $\qquad\square$

The following concentration inequality about vector-valued random variables is commonly used in the literature, see, e.g. Caponnetto & De Vito (2007, Proposition 2) and references therein.

**Lemma D.6.** *Let $H$ be a real separable Hilbert space. Let $\xi, \xi_1, \ldots, \xi_n$ be i.i.d. random variables taking values in $H$. Assume that*

$$\mathbb{E}\|\xi - \mathbb{E}\xi\|_H^m \le \frac{1}{2}m!\sigma^2 L^{m-2}, \forall m = 2, 3, \ldots. \tag{89}$$

*Then for fixed $\delta \in (0, 1)$, one has*

$$\mathbb{P}\left\{\left\|\frac{1}{n}\sum_{i=1}^n \xi_i - \mathbb{E}\xi\right\|_H \le 2\left(\frac{L}{n} + \frac{\sigma}{\sqrt{n}}\right)\ln\frac{2}{\delta}\right\} \ge 1 - \delta. \tag{90}$$

*Particularly, a sufficient condition for (89) is*

$$\|\xi\|_H \le \frac{L}{2} \text{ a.s., and } \mathbb{E}\|\xi\|_H^2 \le \sigma^2.$$

The following Bernstein type concentration inequality about self-adjoint Hilbert-Schmidt operator valued random variable results from applying the discussion in Minsker (2017, Section 3.2) to Tropp (2012, Theorem 7.3.1). It can be found in, e.g., Lin & Cevher (2020, Lemma 24).

**Lemma D.7.** *Let $H$ be a separable Hilbert space. Let $A_1, \ldots, A_n$ be i.i.d. random variables taking values of self-adjoint Hilbert-Schmidt operators such that $\mathbb{E}A_1 = 0$, $\|A_1\| \le L$ almost surely for some $L > 0$ and $\mathbb{E}A_1^2 \preceq V$ for some positive trace-class operator $V$. Then, for any $\delta \in (0, 1)$, with probability at least $1 - \delta$ we have*

$$\left\|\frac{1}{n}\sum_{i=1}^n A_i\right\| \le \frac{2LB}{3n} + \sqrt{\frac{2\|V\|B}{n}}, \quad B = \ln\frac{4\operatorname{Tr} V}{\delta\|V\|}.$$

# E   MORE EXPERIMENTS

In this section we provide more results about the experiments.

## E.1   MORE EXPERIMENTS ON THE INTERVAL

In the following experiments, we use the same setting as in Section 4. We consider other commonly used kernels and set $f^*$ as one of its eigenfunctions. We also compare KRR with another regularization algorithm called spectral cut-off (CUT), which also never saturates like GF (see, e.g. Lin et al. (2018, Example 3.1)).

We introduce another kernel with known explicit forms of eigenfunction, which will be used as the underlying regression function.

**Heavy-side step kernel**   The heavy-side step kernel on $[0, 1]$ is defined by

$$k(x, y) = \min(x, y)(1 - \max(x, y)) = \begin{cases} (1-x)y & \text{if } x \ge y; \\ (1-y)x & \text{if } x < y. \end{cases} \tag{91}$$

The associated RKHS is

$$H^1(0, 1) := \left\{f : [0, 1] \to \mathbb{R} \mid f \text{ is A.C.}, f(0) = f(1) = 0, \int_0^1 (f'(x))^2 \mathrm{d}x < \infty\right\}, \tag{92}$$

with inner product $\langle f, g \rangle_{H^1} := \int_0^1 f'(t)g'(t)\mathrm{d}t$. It is known that the eigen-system of this kernel is

$$\lambda_n = \frac{1}{\pi^2 n^2}, \quad e_n = \sqrt{2}\sin(n\pi x), \quad n = 1, 2, \ldots, \tag{93}$$

and hence $\beta = 0.5$.

We conduct experiments on the two kernels and set the regression function to be one of the eigenfunctions. We report the results in Table E.1 on page 30. The results are generally the same as that of Section 4: GF and CUT methods are similar and they both achieve better performances as $\alpha$ increases, while KRR reaches its best performance at $\alpha = 2$ with resulting max rate approximately 0.8, verifying our theory. We also notice that there are some numerical fluctuation. We attribute them to randomness where we find the deviance is large and numerical error since the eigenvalues are small. In conclusion, the numerical results are supportive.

| Kernel | $\alpha$ | $f^* = e_1$ | | | $f^* = e_2$ | | | $f^* = e_3$ | | |
|---|---|---|---|---|---|---|---|---|---|---|
| | | KRR | GF | CUT | KRR | GF | CUT | KRR | GF | CUT |
| $\min(x, y)$ | 1.5 | .73 | .72 | .73 | .76 | .74 | .73 | .75 | .74 | .75 |
| | 2 | **.76** | .76 | .78 | **.80** | .78 | .79 | **.78** | .78 | .81 |
| | 2.5 | .69 | .80 | .81 | .73 | .81 | .82 | .69 | .82 | .83 |
| | 3 | .58 | .82 | .84 | .62 | .83 | .83 | .59 | .84 | .85 |
| | 3.5 | .49 | **.84** | **.86** | .54 | **.85** | **.88** | .51 | **.86** | **.90** |
| Heavy-side step | 1.5 | .77 | .83 | .71 | .80 | .75 | .73 | **.88** | .75 | .72 |
| | 2 | **.83** | .83 | .74 | **.80** | .80 | .79 | .78 | .80 | .77 |
| | 2.5 | .83 | .87 | .76 | .68 | .84 | .84 | .64 | .84 | .84 |
| | 3 | .75 | .90 | .79 | .57 | .87 | .86 | .54 | .88 | .89 |
| | 3.5 | .65 | **.92** | **.81** | .49 | **.89** | **.88** | .46 | **.92** | **.95** |

Table 2: Convergence rates comparison between KRR, GF and CUT with $\lambda = cn^{-\frac{1}{\alpha+\beta}}$ for various $\alpha$'s. Bold numbers represent the max rate over different choices of $\lambda$.

### E.2 EXPERIMENTS ON THE SPHERE

In this part we conduct experiments beyond dimension 1. We consider some inner-product kernels on the sphere $\mathcal{X} = \mathbb{S}^2 \subset \mathbb{R}^3$ with $\mu$ being the uniform distribution. The reason is that in general it is hard to find an explicit eigen-decomposition of a general kernel, where we can obtain explicit forms of eigen-functions for inner-product kernels on $\mathbb{S}^{d-1}$, which are necessary for us to construct smooth regression functions. These eigen-functions are known as the spherical harmonics, which turn out to be homogeneous polynomials. We refer to Dai & Xu (2013) for a detailed introduction. On $\mathbb{S}^2$, the spherical harmonics are often denoted by $Y_l^m$, $l = 0, 1, 2, \ldots$, $m = -l, \ldots, l$, and $Y_l^m$ is a homogeneous polynomial of order $l$. We pick some of them to be our underlying truth function $f^*$, which are listed below:

$$Y_1^1(x_1, x_2, x_3) = \sqrt{\frac{3}{4\pi}} x_1, \quad Y_2^{-2}(x_1, x_2, x_3) = \frac{1}{2}\sqrt{\frac{15}{\pi}} x_1 x_2,$$

$$Y_3^2(x_1, x_2, x_3) = \frac{1}{4}\sqrt{\frac{105}{\pi}} (x_1^2 - x_2^2) x_3.$$

In terms of kernels, we use the truncated power function $k(x, y) = (1 - \|x - y\|)_+^p$, where $a_+ = \max(a, 0)$. It is known that if $p > \lfloor \frac{d}{2} \rfloor + 1$, this kernel is positive definite on $\mathbb{R}^d$ and thus positive definite on $\mathbb{S}^{d-1}$ (Wendland, 2004, Theorem 6.20). However, we do not know the eigen-decay rate $\beta$ for these kernels.

In the following experiment, we basically follow the same procedure as described in Section 4, except that we choose the regularization parameter by $\lambda = n^{-\theta}$ with various $\theta$. The results are collected in Table E.2 on page 31. We also plot the error curves of one of the experiments in Figure 2 on page 31. The results show that the convergences rates of KRR increase and then decrease as $\theta$ decrease, while the convergences rates of GF keep increasing, and the best convergence rate of KRR is significant slower than that of GF. We conclude that this experiment also justifies the saturation effect and our theory.

| Kernel | $\theta$ | $f^* = Y_1^1$ | | $f^* = Y_2^{-2}$ | | $f^* = Y_3^2$ | |
|---|---|---|---|---|---|---|---|
| | | KRR | GF | KRR | GF | KRR | GF |
| $(1 - \|x - y\|)_+^3$ | 0.6 | .50 | .49 | .49 | .38 | .49 | .49 |
| | 0.4 | **.77** | .70 | .71 | .65 | .70 | .69 |
| | 0.3 | .71 | .80 | **.73** | .74 | **.73** | .78 |
| | 0.2 | .45 | **.96** | .49 | **.83** | .50 | **.89** |
| $(1 - \|x - y\|)_+^4$ | 0.6 | .40 | .37 | .39 | .38 | .39 | .37 |
| | 0.4 | .70 | .65 | .65 | .65 | .65 | .65 |
| | 0.3 | **.77** | .76 | **.74** | .74 | **.74** | .75 |
| | 0.2 | .56 | **.90** | .64 | **.83** | .65 | **.84** |

Table 3: Convergence rates comparison between KRR and GF with $\lambda = cn^{-\theta}$ for various $\theta$'s. Bold numbers represent the max rate over different choices of $\lambda$.

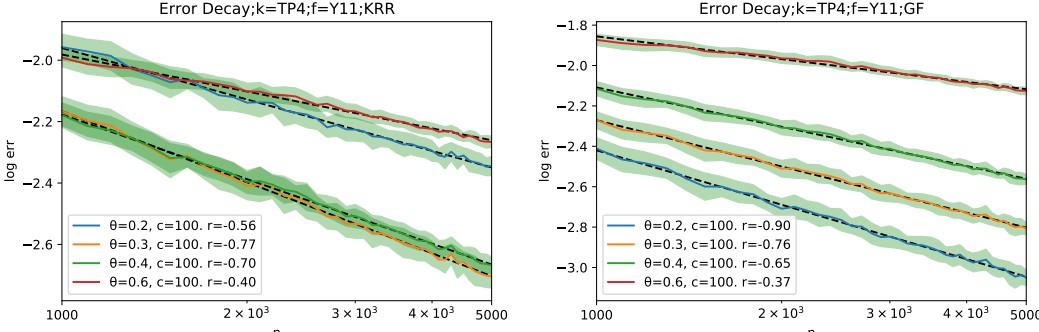

Figure 2: Error decay curves of KRR and GF with kernel $(1 - \|x - y\|)_+^4$ on $\mathbb{S}^2$ and $f^* = Y_1^1$.

