# OpenReview forum: "On the Saturation Effect of Kernel Ridge Regression"
_ICLR.cc/2023/Conference — ICLR 2023 poster_

### Official Review · Reviewer_VSqD · 2022-10-23

**Confidence:** 3
**Correctness:** 3
**Technical Novelty And Significance:** 3
**Empirical Novelty And Significance:** 1
**Recommendation:** 6

**Clarity, Quality, Novelty And Reproducibility:**

A narrative is clear and justified. For me it is hard to judge the originality, because I am not familiar with earlier papers on the saturation effect.

**Strength And Weaknesses:**

I have not checked mathematical proofs completely, though major arguments seems to be correct.

First general question that appears after reading the article is: can the condition (2) be weakened. Does the saturation effect disappear is we have zero noise? E.g. what changes if we set sigma=0 in numerical experiments of Section 4? From the bias-variance decomposition formula on page 4 it seems that the effect should disappear, because the variance term will disappear. Could you comment on that.

**Summary Of The Paper:**

The paper is dedicated to a theoretical analysis of the saturation effect, that is the gap between information theoretical lower bound on the generalization error of KRR and the actual error. This effect has been widely observed in practices. The main result of the paper is given in Theorem 3.1, that deduces a lower bound on the accuracy of KRR under assumptions (A) eigenvalues decay rate, 1) smoothness of kernel and 2) lower bound on the variance of noice.

**Summary Of The Review:**

A paper is theoretical. Statements are justified. Since ICLR is more oriented towards experiments, a paper needs to enhance experimental section. Is everything understood with the saturation effect after that theory was built, or something is still present in experiments that was not covered by theory?

---

> ### Author Response · Authors · 2022-11-17
> **Thank you for your review**
>
> Thank you for your detailed review of our work and your thoughtful comments.
>
>
>
> *"I have not checked mathematical proofs completely, though major arguments seems to be correct. First general question that appears after reading the article is: can the condition (2) be weakened. Does the saturation effect disappear is we have zero noise? E.g. what changes if we set sigma=0 in numerical experiments of Section 4? From the bias-variance decomposition formula on page 4 it seems that the effect should disappear, because the variance term will disappear. Could you comment on that."*
>
> Thank you for your valuable suggestions. Please let us address your concern on condition/assumption (2) first.
>
> The saturation effect proposed in KRR is implicitly assumed that the noise is non-zero. The assumption (2) is actually very weak condition if noise is non-zero. For example, let us take a look of the  classical nonparametric regression models:
>
> ​													$y=f^*(x)+\sigma\epsilon,\quad \epsilon \sim N(0,1),$ where $\epsilon$ is independent of $x$.
>
> It is clear that whenever $\sigma\neq 0$, the the assumption (2) holds automatically.
>
>
>
> We agree that $\sigma=0$ or near 0 is an interesting regime in recent years,
> though it is beyond the scope of the current paper which is devoted to the saturation effect in KRR.
> Historically,  in statistics or machine learning,
> researchers always assume a non-vanishing noise.
> This situation was changed when we try to explain the nice generalization ability of overparametrized neural network.
> The noiseless regime gains lots of attention these years, however, we are faced many technical difficulties.
>
> First, the traditional information-theoretical lower bound does not deal with the noiseless case.
> The convergence of generalization error in noiseless case can in fact be faster than $n^{- \frac{\alpha}{\alpha + \beta}}$.
> A recent work [Cui2021] discusses the generalization error of kernel ridge regression under Gaussian design.
> Their result claims that in the noiseless case, the generalization error rate has an upper bound $n^{- \frac{\min(\alpha,2)}{\beta}}$.
> However, their result is based on a Gaussian design assumption, which turns out to be a very strong assumption.
> We also want to cite [Spigler2020],[Bordelon2020] as two recent works studying the generalization error rate of
> KRR in the noiseless case.
> Both of them give inspiring insight into the generalization error rate.
> We hope these work could clarify your concerns partially.
>
> In addition, we performed a simulation similar as Section 4 to study the saturation effect in the noiseless case.
> We set the variance of noise $\sigma = 0$ and choose the regularization term as zero (interpolation).
> Result shows that although the smoothness of the true function is very smooth,
> the convergence rate of generalization error can not be faster than $n^{-\frac{2}{\beta}}$.
>
>
>
>
>
> *"A narrative is clear and justified. For me it is hard to judge the originality, because I am not familiar with earlier papers on the saturation effect. "*
>
> Thanks for your honesty.  We have provided a paragraph "brief literature review and summary of technical contributions" in the response to the reviewer s1WT
> (due to character limit we cannot put it here).
> We hope the same paragraph could convince you that at least, we have invented some new technical tools to solve a well known open problem.
>
>
>
> *"A paper is theoretical. Statements are justified. Since ICLR is more oriented towards experiments, a paper needs to enhance experimental section. Is everything understood with the saturation effect after that theory was built, or something is still present in experiments that was not covered by theory?"*
>
> We have enhanced our experimental section in the latest revision, considering more truth functions and more kernels.
> We believe that, when the noise is non-zero, we theoretically demonstrated the saturation effect for KRR.
> Our theory is also justified by reasonable amount of numerical experiments.
>
> ### References
>
> [Cui2021] Hugo Cui, Bruno Loureiro, Florent Krzakala, and Lenka Zdeborov’a. Generalization error rates in
> kernel regression: The crossover from the noiseless to noisy regime. In NeurIPS, 2021.
>
> [Bordelon2020] Blake Bordelon, Abdulkadir Canatar, and Cengiz Pehlevan. Spectrum dependent learning curves in
> kernel regression and wide neural networks. In ICML, 2020.
>
> [Spigler2020] Stefano Spigler, Mario Geiger, and Matthieu Wyart. Asymptotic learning curves of kernel methods:
> empirical data versus teacher–student paradigm. Journal of Statistical Mechanics: Theory and
> Experiment, 2020, 2020.

---

### Official Review · Reviewer_iDfT · 2022-10-23

**Confidence:** 3
**Correctness:** 3
**Technical Novelty And Significance:** 4
**Empirical Novelty And Significance:** 3
**Recommendation:** 6

**Clarity, Quality, Novelty And Reproducibility:**

The paper is mostly clear and of high quality. The theoretical contributions are novel.

**Strength And Weaknesses:**

The paper is well-written and easy to follow. The authors provide proof of a long-standing conjecture about the information-theoretical lower bound of KRR. The theoretical contribution is significant.

However, I still have some concerns about the assumptions. The lower bound is obtained under different assumptions than the upper bound. The upper bound (prop 2.2) requires the distributions satisfying conditions (B') and (C), while the proposed lower bound (thm 3.1) requires Assumption 1 and 2. Thus I do not think this paper completely proves the conjecture since it is still unknown whether the proposed lower bound can be obtained under condition (B') and (C).

I'm also confused about the assumptions on the noise. Notice that condition (C) assumes the momentum of the noise is upper bounded but Assumption 2 requires the noise to be non-vanishing. These two assumptions seem somewhat opposite. I hope the authors can provide a detailed discussion of the required assumptions.

minors:

Line 2 of thm 3.1: "satisfies the that" -> "satisfies that"

**Summary Of The Paper:**

This paper studies the saturation effect of KRR. Under some additional assumptions, the paper provides a lower bound of the rate of generalization error for the case where $f\in[\mathcal{H}]^\alpha$ for some $\alpha>2$. The proposed lower bound matches the existing upper bound when $\alpha>2$ and verifies the saturation effect of KRR.

**Summary Of The Review:**

Overall, I think this paper provides an important contribution to the saturation effect of KRR. But the authors need to provide more discussions on the additional assumptions.

---

> ### Author Response · Authors · 2022-11-17
> **Thank you for your review**
>
> Thank you for your review and your thoughtful comments. We appreciate you finding our paper well-written, easy to follow and theoretically contributing.
>
>
>
> *"However, I still have some concerns about the assumptions. The lower bound is obtained under different assumptions than the upper bound. The upper bound (prop 2.2) requires the distributions satisfying conditions (B') and (C), while the proposed lower bound (thm 3.1) requires Assumption 1 and 2. Thus I do not think this paper completely proves the conjecture since it is still unknown whether the proposed lower bound can be obtained under condition (B') and (C).I'm also confused about the assumptions on the noise. Notice that condition (C) assumes the momentum of the noise is upper bounded but Assumption 2 requires the noise to be non-vanishing. These two assumptions seem somewhat opposite. I hope the authors can provide a detailed discussion of the required assumptions."*
>
>
>
> Thanks for that you are interested in our work. We agree that these assumptions are somehow not reader friendly.
> Please let us start with a more friendly example.
> Suppose that we are working with the usual non-parametric regression model:
>
> ​										$y=f^*(x)+\epsilon,\quad \epsilon \sim N(0,\sigma^2),$ where $\epsilon$ is independent of $x$ and $\sigma^2 > 0$,
>
> It is clear that for this model, the **condition (C)** and **assumption 2** hold automatically.
>
> The **assumption 1** is the assumption made on the kernel function (or equivalent, on the corresponding RKHS $\mathcal{H}$)
> and holds for most commonly used kernel functions. e.g., the RBF kernel, the smooth translation invariant kernel, Sobolev kernel, etc.
> Then our claims (thm3.1 and prop 2.2.) hold at least for this model.
> In other words,  we proved the conjectures for a fairly large class of models.
>
> Besides the above example, we now provide more explanations on the **condition (C)** and **condition (B')**
> to clarify your further concerns about our assumptions in the following:
>
> ### Condition (B')
>
> The condition (B') says that $f \in [\mathcal{H}]^\alpha$ and $||f||_{[\mathcal{H}]^\alpha} \leq R$,  where $R$ is some constant.
> The former requirement $f \in [\mathcal{H}]^\alpha$ is already contained in the statement in our main theorem.
> The latter requirement (boundness of $[\mathcal{H}]^\alpha$-norm) is common and necessary in analyzing upper rates since we are consider a class of functions,
> and the constant $R$ will only result in the hidden constant factor  in the upper bound.
> However, in our case of the saturation lower bound, we take an arbitrary fixed regression function,and the boundness is automatically satisfied.
> Therefore, we are essentially considering the same setting as condition (B').
>
> ### Condition (C)
>
> Condition (C) is a moment condition on the noise which requires that the tail probability of the noise must decay fast. This condition is satisfied if the noise is bounded or sub-Gaussian.
> The reason we adopted  this  moment condition instead of the Gaussian noise is to be consistent with previous literatures,
> for example, [Fischer2020, Condition (MOM)], [Lin2018, Assumption 1],  [Caponnetto2007, Hypothesis 2].
>
> We hope that our explanation could clarify your concerns.
>
> ### Minors:
>
> *minors: Line 2 of thm 3.1: "satisfies the that" -> "satisfies that"*
>
> Thanks for your careful reading again. We apologize for our annoying typos.
> We have tried our best to read through the paper and made several modifications in the  new revision.
>
> ## References
>
> [Fischer2020] Simon-Raphael Fischer and Ingo Steinwart. Sobolev Norm Learning Rates for Regularized Least-Squares Algorithms.
> Journal of Machine Learning Research, 21:205:1–205:38.
>
> [Lin2018] Junhong Lin, Alessandro Rudi, L. Rosasco, and V. Cevher.
> Optimal rates for spectral algorithms with least-squares regression over Hilbert spaces.
> Applied and Computational Harmonic Analysis, 48:868–890, 2018.
>
> [Caponnetto2007] Andrea Caponnetto and Ernesto De Vito. Optimal rates for the regularized least-squares algorithm.
> Foundations of Computational Mathematics, 7(3):331–368.

---

### Official Review · Reviewer_1PPr · 2022-10-24

**Confidence:** 3
**Correctness:** 3
**Technical Novelty And Significance:** 3
**Empirical Novelty And Significance:** 3
**Recommendation:** 8

**Clarity, Quality, Novelty And Reproducibility:**

- Novelty: the paper proves a commonly observed conjecture which has impact on the theoretical understanding of KRR.
- Clarity: I found the paper well-written and easy to follow.
- Quality: the theoretical proof looks solid.



**Strength And Weaknesses:**

The paper presents new insights into the saturation phenomenon: the optimally tuned KRR can achieve the minimax rate $n^{-\alpha/(\alpha+\beta)}$ when the smoothness of the underlying truth function is within certain threshold; however, the rate for the estimator is bounded from below by $n^{-2/(2+\beta)}$ if the smoothness exceeds the threshold. This could be a useful contribution to learning theory.

**Summary Of The Paper:**

The paper studies the saturation effect of kernel ridge regression (KRR), providing a proof of the conjecture. The main result is a lower bound on the generalization error which suggests that KRR cannot achieve the information theoretical lower bound when the smoothness of the underground truth function exceeds certain level.

**Summary Of The Review:**

Overall, I think this is an interesting paper with solid theoretical contributions to kernel methods.

---

> ### Author Response · Authors · 2022-11-17
> **Thank you for your review**
>
> Thank you for your positive feedback and for recognizing our contributions to the learning theory. We appreciate your encouragement.
>
> We have updated a new revision with clearer discussion on the assumptions and enhanced experiment section.
> We hope this will make our paper more readable to people who are interested in saturation effects.

---

### Official Review · Reviewer_s1WT · 2022-10-25

**Confidence:** 2
**Correctness:** 3
**Technical Novelty And Significance:** 2
**Empirical Novelty And Significance:** 2
**Recommendation:** 6

**Clarity, Quality, Novelty And Reproducibility:**

In terms of the clarity and novelty, the paper is fine.


**Strength And Weaknesses:**

The paper is partially interesting in terms of considering an unexplored regime where our model is too complex compared with the true prediction function. I am not an expert who can look into the details of the proof, but the derived result seems to provide a clear interpretation and the coincidence with the experiments.

However in practice, the proposed saturation effect can be addressed by changing the kernel functions.


**Summary Of The Paper:**

The authors explain the slow convergence of the Nonparametric Kernel regression with ridge regularization in the regime where the true prediction function is too smooth compared with that of the kernel.

Two parameters have been introduced: although not explicitly explained, \beta controls the span of the kernel in the RKHS, and \alpha controls the smoothness of the true prediction function compared with the used kernel. When \alpha=2, the smoothness is equal to the that of the kernel, and if \alpha > 2, the prediction function is smoother than the used kernel. In this case, the convergence can be slow even though generally the smooth functions are preferred for generalization.

The authors provided an analysis on this regime and showed some experiments confirming the derivation.

**Summary Of The Review:**

The focus of the paper is on the regime where the prediction function is too smooth compared with that of the kernel function. I find the paper  interesting and think the paper can be published at ICLR, but I am not an expert to provide reviews for the quality of the proof techniques.

---

> ### Author Response · Authors · 2022-11-17
> **Thank you for your review**
>
> Thank you for your nice summary of our paper and your valuable feedback.
>
> We agree with you that it would be better to include an explicit description of the parameter $\beta$,
> which controls the span of the RKHS, and $\alpha$,
> which describes the relative smoothness of the regression function with respect to the kernel.
> We have made a modification in the revision, and hope it can help the reader grasp the intuition.
>
> *"However in practice, the proposed saturation effect can be addressed by changing the kernel functions",*
>
> We agree with you that changing the kernel function could improve the performance of kernel ridge regression(KRR).
> Usually, researchers believe that the smoother the truth function, the smaller the generalization error.
> The saturation effect asserts that if one does not choose his kernel carefully,
> the KRR can not provide the optimal regressor when the true regressor function is too smooth.
> This counterintuitive statement provides a theoretical justification on what kind choice of kernel is not favorable to the KRR, i.e.,
> if one plans to perform KRR, then those kernels $k$ (with the corresponding RKHS $\mathcal{H}$ )
> such that $f\in \mathcal{H}^{\alpha}$ for some $\alpha>2$ should not be chosen.
>
>
>
> *"The focus of the paper is on the regime where the prediction function is too smooth compared with that of the kernel function.
> I find the paper interesting and think the paper can be published at ICLR, but I am not an expert to provide reviews for the quality of the proof techniques."*
>
> We are glad to learn that our work is interesting to you.  We understand that it would be hard to judge in short time
> if a proof full of technical details is correct or not.
> Please let us provide a brief review of the history of the saturation effect and a brief summary of
> the key observations and technical contribution below.
>
> **Brief literature review and summary of technical contributions**
>
> The saturation effect is first studied in the area of inverse problems [Neubauer1997, Mathé2004],
> where there is no randomness and the proof is easy.
> In the study of kernel regression (which is also called statistical inverse problem),
> the saturation effect was first proposed in [Bauer2007], but only upper bound was considered and some numerical experiments
> were conducted to illustrate it. Recently, [Dicker2017] made a comparison between KRR and spectral cut-off method,
> showing the convergence rate upper bound of KRR is slower than that of spectral cut-off method.
> However, to the best of our knowledge, no lower bound of KRR has been proved in previous literatures,
> though the phenomenon was widely mentioned in the literature (e.g., in [Lin2020, Lian2021]).
>
> The main technical contributions in our proof consist of the following:
>
> 1. We made a key observation (the Eq. (23) in the revision of the paper) which enables us to handle the variance term in a conceptual way.
> 2. Thanks to the observation 1. We then established a fine-grained approximation so that the error terms are infinitesimal to the main term. This refined approximation is the key step for us to obtain the lower bound of the variance term of KRR. We have to emphasize that in the previous work only concerning the upper bounds, the errors of the same order as the main term is  sufficient.
> 3. In order to fully utilize the concentration inequality of integral operator techniques (e.g., in [Caponnetto2007]), we carefully calculated of the covering number of  regularized basis function family $(T+\lambda)^{-1}k(x,\cdot),~x \in \mathcal{X}$ and employed the empirical process technique (e.g., in [Steinwart2009]).
>
>
> We hope this will convince you that the saturation effect is in lots of researchers' interests and our proof is reliable.
>
>
> ## References
>
> [Neubauer1997] Neubauer, A. (1997).
> On converse and saturation results for Tikhonov regularization of linear ill-posed problems.
>
> [Mathé2004] Mathé, P. (2004).
> Saturation of regularization methods in Hilbert spaces.
>
> [Bauer2007] Bauer, F., Pereverzyev, S., and Rosasco, L. (2007).
> On regularization algorithms in learning theory.
>
> [Dicker2017] Dicker, L., Foster, D. P., and Hsu, D. J. (2017).
> Kernel ridge vs. principal component regression: Minimax bounds and the qualification of regularization operators.
>
> [Caponnetto2007] Andrea Caponnetto and Ernesto De Vito.(2007). Optimal rates for the regularized least-squares algorithm.
>
> [Steinwart2009] Steinwart, I., Hush, D., and Scovel, C. (2009).
> Optimal rates for regularized least squares regression.
>
> [Lin2020] Junhong Lin and Volkan Cevher. (2020). Optimal convergence for distributed learning with stochastic gradient methods
> and spectral algorithms.
>
> [Lian2021] Heng Lian, Jiamin Liu, and Zengyan Fan. (2021).
> Distributed learning for sketched kernel regression.

---

### Decision · Program_Chairs · 2023-01-20

**Decision:**

Accept: poster

**Justification For Why Not Higher Score:**

The impact of the paper for the ICLR community, more on the empirical and deep learning side, could be limited.

**Justification For Why Not Lower Score:**

The technical quality of the production is solid, solving a longstanding conjecture.

**Metareview: Summary, Strengths And Weaknesses:**

The paper studies the so-called saturation effect of kernel ridge regression (KRR), ie, the inability of KRR to achieve proper learning when the smoothness of the underground truth function exceeds certain level. This is a theoretical paper addressing and answering a longstanding conjecture on this saturation effect.

+ technically clear and sound paper
+ solves a conjecture important on KRR, a widely used ML learning algorithm
- write-up: there are many typos to be corrected




**Note From Pc:**

if the above contains the word "oral" or "spotlight" please see: "oral" presentation means -> notable-top-5% and "spotlight" means -> notable-top-25%. As stated in our emails, we are disassociating presentation type from AC recommendations